EMBO
Molecular Medicine

# Myeloid-resident neuropilin-1 promotes choroidal neovascularization while mitigating inflammation

Elisabeth M M A Andriessen[1,†], François Binet[2,3,†], Frédérik Fournier[4], Masayuki Hata[4], Agnieszka Dejda[3], Gaëlle Mawambo[4], Sergio Crespo-Garcia[3,4] (iD), Frédérique Pilon[3], Manuel Buscarlet[4], Karine Beauchemin[2], Véronique Bougie[2], Garth Cumberlidge[2], Ariel M Wilson[3], Steve Bourgault[6], Flavio A Rezende[3], Normand Beaulieu[2], Jean-Sébastien Delisle[5] & Przemyslaw Sapieha[1,2,3,4,*,†] (iD)

## Abstract

**Age-related macular degeneration (AMD) in its various forms is a leading cause of blindness in industrialized countries. Here, we provide evidence that ligands for neuropilin-1 (NRP1), such as Semaphorin 3A and VEGF-A, are elevated in the vitreous of patients with AMD at times of active choroidal neovascularization (CNV). We further demonstrate that NRP1-expressing myeloid cells promote and maintain CNV. Expression of NRP1 on cells of myeloid lineage is critical for mitigating production of inflammatory factors such as IL6 and IL1β. Therapeutically trapping ligands of NRP1 with an NRP1-derived trap reduces CNV. Collectively, our findings identify a role for NRP1-expressing myeloid cells in promoting pathological angiogenesis during CNV and introduce a therapeutic approach to counter neovascular AMD.**

**Keywords** age-related macular degeneration; angiogenesis; inflammation; mononuclear phagocytes; neuropilin-1

**Subject Category** Vascular Biology & Angiogenesis

## Introduction

Age-related macular degeneration (AMD) is a slowly progressing condition of the aging eye and the leading cause of central vision loss in industrialized countries (Friedman *et al*, 2004; Rein *et al*, 2006; Jonas *et al*, 2014; Wong *et al*, 2014). Central vision loss from AMD poses a significant burden on health care systems and profoundly impacts well-being and mental health (Taylor *et al*, 2016). In the early asymptomatic stages of disease, insoluble extracellular lipid aggregates termed drusen accumulate in the subretinal space. Inadequate clearance of these deposits can trigger a pathologic inflammatory response and ensuing tissue damage (Guillonneau *et al*, 2017; Handa *et al*, 2017). Advanced AMD is often classified into "dry" atrophic AMD or "wet" neovascular (NV) AMD. Visual impairment in the late stages of atrophic AMD is characterized by areas with progressive degeneration of the retinal pigment epithelium (RPE) and the photoreceptors that rely on RPE for support (Ambati & Fowler, 2012; Lim *et al*, 2012). While vision loss resulting from the dry form of AMD is typically gradual and protracted, wet AMD can rapidly compromise the central visual field. This typically occurs when neovascularization from the choroid (choroidal neovascularization; CNV) sprouts into the subretinal space and neuro-retina, hemorrhages, leaks and ultimately provokes photoreceptor death, fibrovascular scarring and retinal detachment of the macular region (Lim *et al*, 2012).

To date, the mechanisms that precipitate NV AMD remain only partially defined, with vascular endothelial growth factor A (VEGF-A) playing a cardinal role in CNV (Ambati & Fowler, 2012). Current standards of care for wet AMD such as Aflibercept, Ranibizumab, and off-label Bevacizumab target VEGF-A and have revolutionized treatment of NV AMD. Unfortunately, long-term use of anti-VEGF-A therapies may have limited efficacy (Comparison of Age-related Macular Degeneration Treatments Trials Research Group *et al*, 2016), possible neuronal side-effects (Robinson *et al*, 2001), and have been shown to cause degeneration of the RPE-choriocapillaris complex in mouse models (Saint-Geniez *et al*, 2008; Kurihara *et al*, 2012; Berber *et al*, 2017). In addition, approximately 1 out of 10 treated patients does not respond to anti-VEGF-A therapy (Krebs *et al*, 2013) with NV network complexes persisting despite monthly intravitreal injections. Therefore, alternative treatments that block retinal neovascularization in AMD are required. A pharmacogenomic link has been suggested with the variability in treatment

1   Department of Biomedical Sciences, University of Montreal, Montreal, QC, Canada
2   SemaThera Inc., Montreal, QC, Canada
3   Department of Ophthalmology, University of Montreal, Montreal, QC, Canada
4   Department of Biochemistry and Molecular Medicine, University of Montreal, Montreal, QC, Canada
5   Department of Medicine, Maisonneuve-Rosemont Hospital Research Centre, University of Montreal, Montreal, QC, Canada
6   Department of Chemistry, Université du Québec à Montréal, Montreal, QC, Canada
    *Corresponding author. Tel: +1 514 252 3400; Fax: +1 514 252 3430; E-mail: mike.sapieha@umontreal.ca
    †These authors contributed equally to this work

response (Lazzeri *et al*, 2013; Lores-Motta *et al*, 2018), and it has recently been proposed that a single-nucleotide polymorphism (SNP) in Neuropilin-1 (*Nrp1*; rs2070296) is associated with decreased anti-VEGF-A therapy response (Lores-Motta *et al*, 2016).

NRP1 is a single-pass transmembrane receptor with a large ~860 amino acid extracellular domain subdivided into 3 sub-domains: a large extracellular domain with two CUB motifs (A1, A2), two domains with similarity to coagulation factor V/VIII (B1, B2), a MAM domain (C), and a single transmembrane domain (TM) followed by a short cytoplasmic domain (CD; Geretti *et al*, 2008). The A1, A2 and B1 domains bind Semaphorin 3A (SEMA3A) while the B1 and B2 domains bind VEGF-A, transforming growth factor beta (TGF-β), placental growth factor 2 (PGF) (Mamluk *et al*, 2002), and platelet-derived growth factor (PDGF; Antipenko *et al*, 2003; Raimondi *et al*, 2016; Muhl *et al*, 2017; Miyauchi *et al*, 2018). NRP1 can collaborate with several receptors and their ligands such as VEGFR2 and VEGF-A (Soker *et al*, 1998; Soker *et al*, 2002), Plexin A1 and SEMA3A (Takahashi *et al*, 1999), TGF-βR and TGF-β (Glinka *et al*, 2011), and PDGF-R and PDGF-BB (Ball *et al*, 2010) and hence has the potential to modulate multiple receptor signaling pathways (Nakamura & Goshima, 2002; Prud'homme & Glinka, 2012; Rizzolio, 2017). Of note, all above ligands are known to regulate angiogenesis, suggesting that NRP1-mediated signaling could be of interest for NV AMD (Battegay *et al*, 1994; Massague *et al*, 2000; Hoeben *et al*, 2004; Acevedo *et al*, 2008; Funasaka *et al*, 2014).

In addition to vessels and neurons, NRP1 is highly expressed on cells of both the innate and adaptive immune system (Roy *et al*, 2017) where it plays an important role in homing and modulating myeloid cell function (Takamatsu & Kumanogoh, 2012; Casazza *et al*, 2013; Roy *et al*, 2017; Wilson *et al*, 2018) and in T-cell migration (Lepelletier *et al*, 2007) and differentiation (Renand *et al*, 2013). Here, we sought to elucidate the contribution of myeloid-resident NRP1 in NV AMD.

## Results

### NRP1 ligands are elevated in patients with NV AMD and in a mouse model of CNV

NRP1 has been implicated in diseases characterized by deregulated vasculature such as cancer (Ellis, 2006; Rizzolio & Tamagnone, 2011; Jubb *et al*, 2012; Prud'homme & Glinka, 2012; Chaudhary *et al*, 2014), and in diseases of the retina such as proliferative

diabetic retinopathy (Dejda *et al*, 2014), diabetic macular edema (Cerani *et al*, 2013; Sodhi *et al*, 2019) and retinopathy of prematurity (Joyal *et al*, 2011; Dejda *et al*, 2014). We therefore sought to determine whether NRP1 ligands were present in NV AMD. We obtained vitreous from patients diagnosed with active proliferating NV AMD as determined by fundus imaging and optical coherence tomography (OCT) and from age- and sex-matched control patients with non-vascular retinal pathologies such as macular hole or epiretinal membranes. Representative horizontal B-scan and thickness maps of an AMD patient with active CNV lesions and a control patient with a non-vascular retinal pathology (macular hole) are shown in Fig 1A and B. Detailed characteristics of patients are included in Table 1. We used ELISA-based detection for NRP1 ligands and found a significant increase in VEGF-A, from 129 to 513 ng/ml and SEMA3A, from 0.001 to 0.33 ng/ml (Fig 1C and D). TGF-β showed a trend toward increase from 1 ng/ml in control patients to 1.3 ng/ml in NV AMD patients (Fig 1E), while PDGF-BB and PGF did not change (Fig 1F and G) as previously reported (Mimura *et al*, 2019).

We next modeled NV AMD in mice by subjecting them to the laser-induced photocoagulation model of CNV, where disruption of the Bruch's membrane triggers sprouting of subretinal vessels from the choroid (Lambert *et al*, 2013) as depicted with fundus photography and fluorescein angiography (Fig 1H). Although this model does not mimic chronic aspects of human disease, it is relatively reproducible and widely used to model CNV in NV AMD. Following induction of CNV, we sacrificed mice and collected RPE-choroid-sclera complexes over the 2-week period of active CNV (3, 7, 10, or 14 days after laser burn). We assessed transcript levels of *Nrp1* ligand by real-time quantitative PCR (RT-qPCR) and *Vegfa* levels were found to rise significantly compared with naïve choroids at day 10 (Fig 1I), while *Sema3a* levels rose twofold by day 14 (Fig 1J). *Tgfb1* transcripts increased as early as 3 days after laser and were elevated throughout the course until day 14 (Fig 1K). Consistent with PDGF-BB levels in human vitreous (Fig 1F), levels of *Pdgfb* did not significantly vary (Fig 1L). *Pgf* expression dropped in the first week after burn and rose slightly around day 10 (Fig 1M) as reported by others (Crespo-Garcia *et al*, 2017). Collectively, these data obtained in both humans and mice suggest that various ligands of NRP1 are elevated in NV AMD.

### NRP1-expressing mononuclear phagocytes rise in the retina upon injury and promote CNV

Under physiological conditions, the subretinal space and photoreceptor cell layer are devoid of mononuclear phagocytes (Guillonneau

---

**Figure 1. NRP1 ligands are elevated in patients with NV AMD and in a mouse model of CNV.**

A  Optical coherence tomography (OCT) horizontal B-scan and thickness map of neovascular age-related macular degeneration (NV AMD) patient with active lesions (1), cystoid macular edema (2), subretinal fibrin deposits (3), and serous retinal detachment (4).
B  Optical coherence tomography (OCT) horizontal B-scan and thickness map of control patient with a medium sized, stage 3, full thickness macular hole (5).
C–G  Vitreous humor analyzed by ELISA for VEGF-A (C); n = 7 (Ctrl), 9 (NV AMD), SEMA3A (D); n = 10 (Ctrl), 10 (NV AMD), TGFβ (E); n = 8 (Ctrl), 7 (NV AMD), PDGF-BB (F); n = 5 (Ctrl),6 (NV AMD), PGF (G): n = 5 (Ctrl), 5 (NV AMD). Dots represent concentrations of individual patient samples.
H  Micron IV infrared and fluorescein in vivo imaging of naïve mouse fundus and following laser-induced CNV at D0, D3, D7, D14.
I–M  Time course of mRNA expression of NRP1 ligands in mouse RPE-choroid-sclera complexes relative to naïve (no burn), 3 (D3), 7 (D7), 10 (D10), and 14 (D14) days after burn for *Vegfa* (I); n = 11 (No burn), 6 (D3), 7 (D7), 4 (D10 and D14), *Sema3a* (J); n = 17 (No burn), 6 (D3), 4 (D7), 3 (D10 and D14), *Tgfb1* (K); n = 14 (No burn), 3 (D3), 7 (D7), 3 (D10), 6 (D14), *Pdgfb* (L); n = 9 (No burn), 5 (D3 and D7), 4 (D10 and D14), *Pgf* (M); n = 6 (No burn), 4 (D3), 3 (D7), 6 (D10), 3 (D14).

Data information: All comparisons between groups were analyzed using a Student's unpaired *t*-test (C-G) or a one-way analysis of variance (ANOVA) and Dunnett's multiple comparisons test (I-M); *P < 0.05, **P < 0.01, ***P < 0.001; error bars represent mean ± SEM; exact *P* values listed in Appendix Table S1.

---

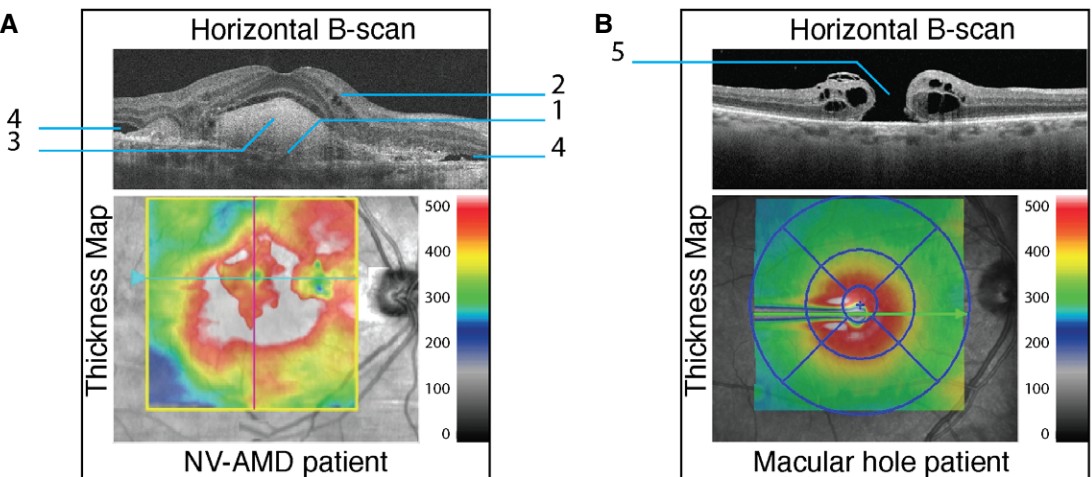

1. Active lesions, 2. Cystoid macular edema, 3. Subretinal fibrin deposits, 4. Serous retinal detachment, 5. Macular hole

**Figure 1.**

**Table 1. Characteristics of donors of vitreous samples used in Fig 1C–G**

| Pathology | ELISA | Age |
|-----------|-------|-----|
| Control Patients | | |
| MH | VEGF, S3A, TGF, PDGF | – |
| ERM | VEGF, S3A, TGF, PDGF | 72 |
| MH | VEGF, S3A, TGF, PGF | 82 |
| ERM | VEGF, S3A, TGF | 57 |
| ERM | VEGF, S3A, TGF | 93 |
| ERM | VEGF, S3A | 76 |
| ERM | VEGF, PGF | 82 |
| MH | S3A, TGF, PDGF | 66 |
| ERM | TGF, PDGF | 54 |
| MH | TGF, PDGF | 65 |
| MH | S3A | 71 |
| ERM | S3A | 71 |
| ERM | S3A, PGF | 64 |
| ERM | PGF | 81 |
| ERM | PGF | 80 |
| *Age Mean ± SEM; VEGF: 77 ± 4.9; S3A: 72 ± 3.5; TGF: 69 ± 5.2; PDGF: 64 ± 3.7; PGF: 78 ± 3.5* | | |
| NV AMD Patients | | |
| NV AMD | VEGF, S3A, TGF, PDGF | 96 |
| NV AMD | VEGF, S3A, TGF, PDGF | 79 |
| NV AMD | VEGF, S3A, TGF, PDGF | 86 |
| NV AMD | VEGF, S3A, TGF, PDGF | 75 |
| NV AMD | VEGF, S3A, TGF, PDGF | 76 |
| NV AMD | VEGF, TGF, PDGF | 80 |
| NV AMD | VEGF, S3A | 84 |
| NV AMD | VEGF | 80 |
| NV AMD | VEGF | 74 |
| NV AMD | S3A, TGF, PDGF, PGF | 79 |
| NV AMD | S3A, PGF | 91 |
| NV AMD | S3A | 84 |
| NV AMD | S3A, PGF | 80 |
| NV AMD | PGF | 71 |
| NV AMD | PGF | 85 |
| *Age Mean ± SEM; VEGF: 81 ± 2.3; S3A: 83 ± 2.1; TGF: 82 ± 2.7; PDGF: 82 ± 2.7.; PGF: 81 ± 3.3* | | |

MH, Macular hole; ERM, Epiretinal membrane; NV AMD, Neovascular AMD.

*et al*, 2017). In late AMD, the immunosuppressive subretinal environment is disturbed and mononuclear phagocytes accumulate and contribute to pathogenesis (Langmann, 2007; Ambati *et al*, 2013; Guillonneau *et al*, 2017; Rashid *et al*, 2019). Secondary to laser-induced CNV, mononuclear phagocytes including microglia and circulating monocytes are recruited to sites of neovascularization (Yu *et al*, 2020). NRP1 is highly expressed on retinal mononuclear phagocytes (Dejda *et al*, 2016), and we have previously

demonstrated a role for NRP1-expressing myeloid cells in mediating pathological angiogenesis in oxygen-induced retinopathy (Dejda *et al*, 2014; Dejda *et al*, 2016). Hence, we sought to determine the contribution of NRP1-expressing mononuclear phagocytes to CNV.

Analysis by FACS of whole retinas and RPE-choroid-sclera complexes at day 3 (D3) of CNV revealed a rise in Ly6G$^-$, F4/80$^+$, CD11b$^+$ mononuclear phagocytes in laser-burned eyes when compared to controls (Fig 2A and B). Importantly, we observed a proportional threefold increase in NRP1$^+$ mononuclear phagocytes at D3 when compared to non-laser eyes (Fig 2C) (gating scheme and internal controls in Fig EV1A–C). In order to establish the role of mononuclear phagocyte-resident NRP1 on CNV, we generated a myeloid-specific knockout of *Nrp1* by crossing *Nrp1*-floxed mice with LysM-Cre (LysM-Cre/*Nrp1*$^{+/+}$) mice, yielding LysM-Cre/*Nrp1*$^{fl/fl}$ offspring. The resulting mice showed a robust reduction in *Nrp1* transcript (Fig 2D) and protein (Fig 2E and F) expression in bone marrow-derived macrophages (BMDM) when compared to LysM-Cre/*Nrp1*$^{+/+}$ littermate controls. FACS analysis revealed a ~30% reduction in NRP1$^+$ microglia in the retina and RPE-choroid-sclera complexes before and 3 days after laser (Fig EV1D and E). LysM-Cre/*Nrp1*$^{fl/fl}$ mice on regular diets did not show any difference in body weight, size, or open-field activity when compared to littermates. Immunofluorescence of lesion sites on flat-mounted RPE-choroid-sclera complexes at D3 post-laser burn reveal that NRP1-positive mononuclear phagocytes (labeled with IBA1) are recruited to sites of injury (Fig 2G) while, as expected, LysM-Cre/*Nrp1*$^{fl/fl}$ mice did not show accretion of NRP1-expressing mononuclear phagocytes.

Throughout the course of CNV, prior to laser burn until D14 post-laser burn, the number of mononuclear phagocytes present in RPE-choroid-sclera complexes of either LysM-Cre/*Nrp1*$^{fl/fl}$ or control LysM-Cre/*Nrp1*$^{+/+}$ mice followed similar trends. Analysis by FACS or immunofluorescence revealed that mononuclear phagocytes in either LysM-Cre/*Nrp1*$^{fl/fl}$ or control LysM-Cre/*Nrp1*$^{+/+}$ mice remained similar over time and did not show significant difference either prior to laser burn or at D3, D7, or D14 (Figs 2H–J and EV1F and G).

Two weeks after laser burn (D14), quantification of compressed Z-stack confocal images of FITC-dextran-perfused neovessels revealed a ~30–40% decrease in CNV in LysM-Cre/*Nrp1*$^{fl/fl}$ mice compared with controls (Fig 2K–N). The average size of isolectin B4 (IB4)-labeled impact areas did not differ between groups (Fig 2 M) suggesting that the observed effect is directly on nascent vasculature. Interestingly, the extent of CNV did not vary between groups in the first week post-laser burn (Fig EV1H–K) implying that NRP1-expressing mononuclear phagocytes partake in later stages of disease. Together, these data suggest that while levels of mononuclear phagocytes are similar between LysM-Cre/*Nrp1*$^{+/+}$ and LysM-Cre/*Nrp1*$^{fl/fl}$ during CNV, NRP1-expressing mononuclear phagocytes promote and maintain neovascularization in the later stages of CNV.

## NRP1-expressing mononuclear phagocytes display a pro-angiogenic phenotype

Given that equal numbers of mononuclear phagocytes are present in the back of the eye of LysM-Cre/*Nrp1*$^{+/+}$ and LysM-Cre/*Nrp1*$^{fl/fl}$ mice following laser burn, yet NRP1$^+$ mononuclear phagocytes

promote CNV (Fig 2), we sought to determine the impact of loss of myeloid-resident NRP1 on choroidal inflammation during CNV. Three days after laser burn, we observed a significant rise in mRNA transcripts for interleukin 1 β (*Il1b*) and interleukin 6 (*Il6*) (Fig 3A and B) in RPE-choroid complexes of LysM-Cre/*Nrp1*^fl/fl mice when compared to controls, with a ~twofold increase in *Il1b* and ~sevenfold increase in *Il6*. No significant change was detected in *Vegfa* or tumor necrosis factor (*Tnf*) (Fig 3C and D). A similar pattern of expression was observed seven days post-laser burn (Fig 3E–H).

We next investigated immune activation in bone marrow-derived macrophages (BMDM) from LysM-Cre/*Nrp1*^+/+ and LysM-Cre/*Nrp1*^fl/fl mice. BMDMs were derived from bone marrow cells using macrophage colony-stimulating factor (M-CSF) which can induce a predominantly anti-inflammatory M2-like cell population (Fleetwood *et al*, 2007; Lacey *et al*, 2012). However, we observed a significant increase in NF-κB p65 phosphorylation in NRP1-deficient BMDMs (Fig 3I and J), accompanied by increased levels of *Il1b* and *Il6* (Fig 3K and L) indicative of classical activation of

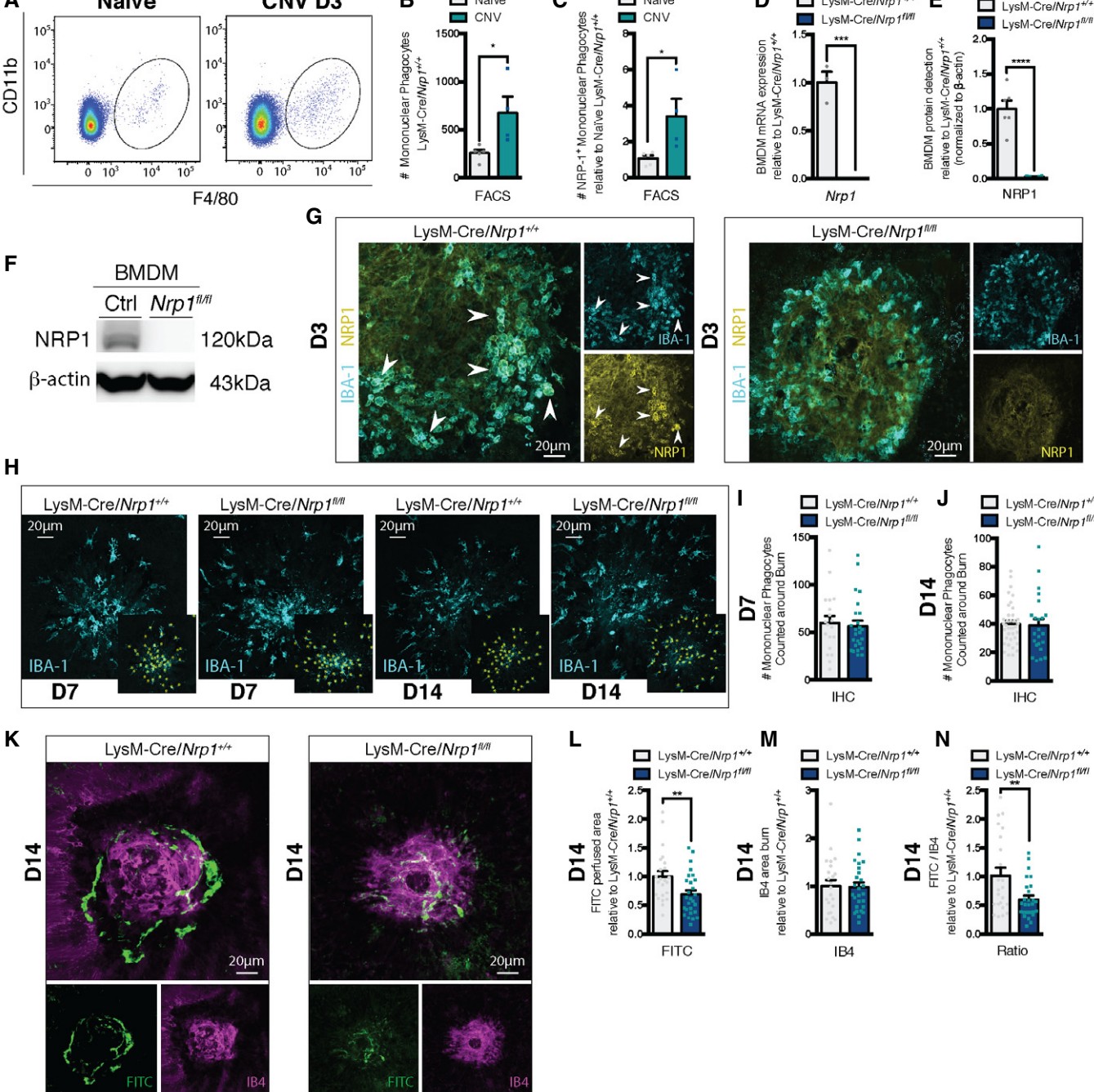

**Figure 2.**

◄

**Figure 2. NRP1-expressing mononuclear phagocytes increase in the retina upon injury and promote CNV.**

A    Representative FACS plots of retinas and sclera-choroid-RPE cell complexes from naïve and burned mice 3 days (D3) after laser burn.
B    Quantification of mononuclear phagocytes (Ly6G⁻, F4/80⁺, CD11b⁺) in retinas and sclera-choroid-RPE cell complexes at D3 relative to naïve; *n* = 5 (Naïve), 4 (CNV).
C    Quantification of NRP1⁺ mononuclear phagocytes (Ly6G⁻, F4/80⁺, CD11b⁺, NRP1⁺) in retinas and sclera-choroid-RPE cell complexes at D3 relative to naïve; *n* = 5 (Naïve), 4 (CNV).
D    mRNA expression of *Nrp1* in BMDM relative to LysM-Cre/*Nrp1*$^{+/+}$; *n* = 3 (LysM-Cre/*Nrp1*$^{+/+}$), *n* = 4 (LysM-Cre/*Nrp1*$^{fl/fl}$).
E    Quantification of NRP1 protein expression in LysM-Cre/*Nrp1*$^{+/+}$ and LysM-Cre/*Nrp1*$^{fl/fl}$ BMDM; *n* = 6.
F    Representative Western blot showing NRP1 expression in LysM-Cre/*Nrp1*$^{+/+}$ (Ctrl) and LysM-Cre/*Nrp1*$^{fl/fl}$ (Nrp1$^{fl/fl}$).
G    Representative confocal images of NRP1 and IBA1-stained mononuclear phagocytes on choroidal flat mounts from LysM-Cre/*Nrp1*$^{+/+}$ and LysM-Cre/*Nrp1*$^{fl/fl}$ mice at D3. Arrowheads indicate NRP1-positive mononuclear phagocytes. Scale bar: 20 μm.
H    Representative confocal images of IBA-1-stained mononuclear phagocytes on choroidal flat mounts from LysM-Cre/*Nrp1*$^{+/+}$ and LysM-Cre/*Nrp1*$^{fl/fl}$ mice at D7 and D14. Examples of macrophage quantification (yellow stars) are presented in side panels. Scale bar: 20 μm.
I, J    Total number of IBA-1-positive mononuclear phagocytes counted around laser impact area on confocal images of choroidal flat mounts at D7 (I) and D14 (J); *n* = 19 burns (D7 LysM-Cre/*Nrp1*$^{+/+}$), *n* = 25 burns (D7 LysM-Cre/*Nrp1*$^{fl/fl}$), *n* = 37 burns (D7 LysM-Cre/*Nrp1*$^{+/+}$), *n* = 23 burns (D7 LysM-Cre/*Nrp1*$^{fl/fl}$), 3–5 mice with ~4 burns per eye.
K    Compilation of representative compressed Z-stack confocal images of FITC–dextran-labeled CNV and IB4-stained laser impact area from LysM-Cre/*Nrp1*$^{+/+}$ and LysM-Cre/*Nrp1*$^{fl/fl}$ mice at D14. Scale bar: 20 μm.
L–N    Quantification of area of FITC–dextran-labeled CNV (L), isolectin B4 (IB4)-stained laser impact area (M) and the ratio of FITC/IB4 per laser burn (N) relative to LysM-Cre/*Nrp1*$^{+/+}$ at D14; *n* = 23 burns (LysM-Cre/*Nrp1*$^{+/+}$), *n* = 27 burns ( LysM-Cre/*Nrp1*$^{fl/fl}$).

Data information: All comparisons between groups were analyzed using a Student's unpaired *t*-test; *P < 0.05, **P < 0.01, ***P < 0.001, ****P < 0.0001; error bars represent mean ± SEM; exact *P* values listed in Appendix Table S1.

mononuclear phagocytes and suggesting that the absence of NRP1 in myeloid cells led to increased levels of these cytokines in RPE-choroid-sclera complexes (Fig 3A and B). In line with these findings, FACS analysis revealed a ~twofold increase in M1-like (F4/80⁺, CD11b⁺, CD11c⁺, CD206⁻) BMDMs in LysM-Cre/*Nrp1*$^{fl/fl}$ mice (Fig 3M and N) (gating scheme in Fig EV2), accompanied by a ~25% decrease in M2-like (F4/80⁺, CD11b⁺, CD11c⁻, CD206⁺) cells (Fig 3M and O).

Moreover, transcriptomic analysis by RNA sequencing (RNA-seq) and gene set enrichment analysis (GSEA) of NRP1-deficient peritoneal macrophages revealed a significant decrease in transcripts from the GO Angiogenesis gene set (Fig 3P) in LysM-Cre/*Nrp1*$^{fl/fl}$ macrophages when compared to wild-type controls. These data support the notion that the absence of NRP1 on mononuclear phagocytes renders them less pro-angiogenic and more pro-inflammatory. These findings are consistent with the lower levels of CNV observed in LysM-Cre/*Nrp1*$^{fl/f}$ mice (Fig 2K–N) and other studies suggesting that heightened inflammation may reduce CNV in the laser-induced mouse model (Zandi *et al*, 2015).

**Therapeutic intravitreal administration of soluble NRP1 reduces CNV in mice**

Based on the above data, we sought to determine the therapeutic value of interfering with NRP1 ligands on the outcome of CNV. We generated a recombinant NRP1-derived trap consisting of the extracellular domain of NRP1 (Fig 4A). This trap binds and neutralizes NRP1 ligands (Cerani *et al*, 2013; Dejda *et al*, 2014). We initially confirmed the binding of the NRP1 ligands SEMA3A and VEGF-A to the trap by surface plasmon resonance (SPR). The trap was covalently immobilized by standard amine coupling on a carboxymethyl dextran sensor chip with lower charge (CM4) and ligands were injected at various concentrations to evaluate binding kinetics in real-time. Sensorgrams revealed fast association and low dissociation for both ligands, which followed a simple 1:1 stoichiometry model. Equilibrium dissociation constants ($K_d$) of 0.67 nM and 4.32 nM were, respectively, extracted for SEMA3A and VEGF-A (Fig 4B–D), indicative of high and physiologically relevant binding affinities. Interestingly, while the association rate constants were somewhat

**Figure 3. NRP1-expressing mononuclear phagocytes display a pro-angiogenic alternatively activated phenotype.**                                          ►

A–D    mRNA expression of inflammation markers relative to LysM-Cre/*Nrp1*$^{+/+}$ in mouse RPE-choroid-sclera complexes at D3 for *Il1b* (A); *n* = 10 (LysM-Cre/*Nrp1*$^{+/+}$ and LysM-Cre/*Nrp1*$^{fl/fl}$), *Il6* (B); *n* = 7 (LysM-Cre/*Nrp1*$^{+/+}$ and LysM-Cre/*Nrp1*$^{fl/fl}$), *Vegfa* (C); *n* = 10 (LysM-Cre/*Nrp1*$^{+/+}$ and LysM-Cre/*Nrp1*$^{fl/fl}$), *Tnf* (D); *n* = 5 (LysM-Cre/*Nrp1*$^{+/+}$), *n* = 10 (LysM-Cre/*Nrp1*$^{fl/fl}$).
E–H    mRNA expression of inflammation markers relative to LysM-Cre/*Nrp1*$^{+/+}$ in mouse RPE-choroid-sclera complexes at D7 for *Il1b* (E); *n* = 6 (LysM-Cre/*Nrp1*$^{+/+}$), *n* = 10 (LysM-Cre/*Nrp1*$^{fl/fl}$), *Il6* (F); *n* = 3 (LysM-Cre/*Nrp1*$^{+/+}$), *n* = 4 (LysM-Cre/*Nrp1*$^{fl/fl}$), *Vegfa* (G); *n* = 6 (LysM-Cre/*Nrp1*$^{+/+}$), *n* = 10 (LysM-Cre/*Nrp1*$^{fl/fl}$), *Tnf* (H); *n* = 8 (LysM-Cre/*Nrp1*$^{+/+}$), *n* = 12 (LysM-Cre/*Nrp1*$^{fl/fl}$).
I    Representative Western blot showing pNF-κB expression in LysM-Cre/*Nrp1*$^{+/+}$ (Ctrl) and LysM-Cre/*Nrp1*$^{fl/fl}$ (Nrp1$^{fl/fl}$).
J    Quantification of pNF-κB expression in LysM-Cre/*Nrp1*$^{+/+}$ and LysM-Cre/*Nrp1*$^{fl/fl}$ BMDM; *n* = 6.
K, L    mRNA expression relative to LysM-Cre/*Nrp1*$^{+/+}$ of inflammation markers in mouse BMDMs for *Il1b* (K); *n* = 4 (LysM-Cre/*Nrp1*$^{+/+}$), *n* = 5 (LysM-Cre/*Nrp1*$^{fl/fl}$) and *Il6* (L); *n* = 3 (LysM-Cre/*Nrp1*$^{+/+}$), *n* = 2 (LysM-Cre/*Nrp1*$^{fl/fl}$).
M    Representative FACS plots of M1 and M2-Like macrophages in LysM-Cre/*Nrp1*$^{+/+}$ and LysM-Cre/*Nrp1*$^{fl/fl}$ BMDMs.
N, O    Quantification of M1-Like macrophages (F4/80⁺, CD11b⁺, CD11c⁺, CD206⁻)(N), M2-Like macrophages (F4/80⁺, CD11b⁺, CD11c⁻, CD206⁺) (O) in LysM-Cre/*Nrp1*$^{+/+}$ and LysM-Cre/*Nrp1*$^{fl/fl}$ BMDMs relative to LysM-Cre/*Nrp1*$^{+/+}$; *n* = 5.
P    Heatmap (left) and enrichment plot (right) of GO Angiogenesis gene set enrichment analysis (GSEA) of wild-type and LysM-Cre/*Nrp1*$^{fl/fl}$ peritoneal macrophages; *n* = 2. NES, normalized enrichment score; FDR, false discovery rate.

Data information: All comparisons between groups were analyzed using a Student's unpaired *t*-test; *P < 0.05, **P < 0.01; error bars represent mean ± SEM; exact *P* values listed in Appendix Table S1.

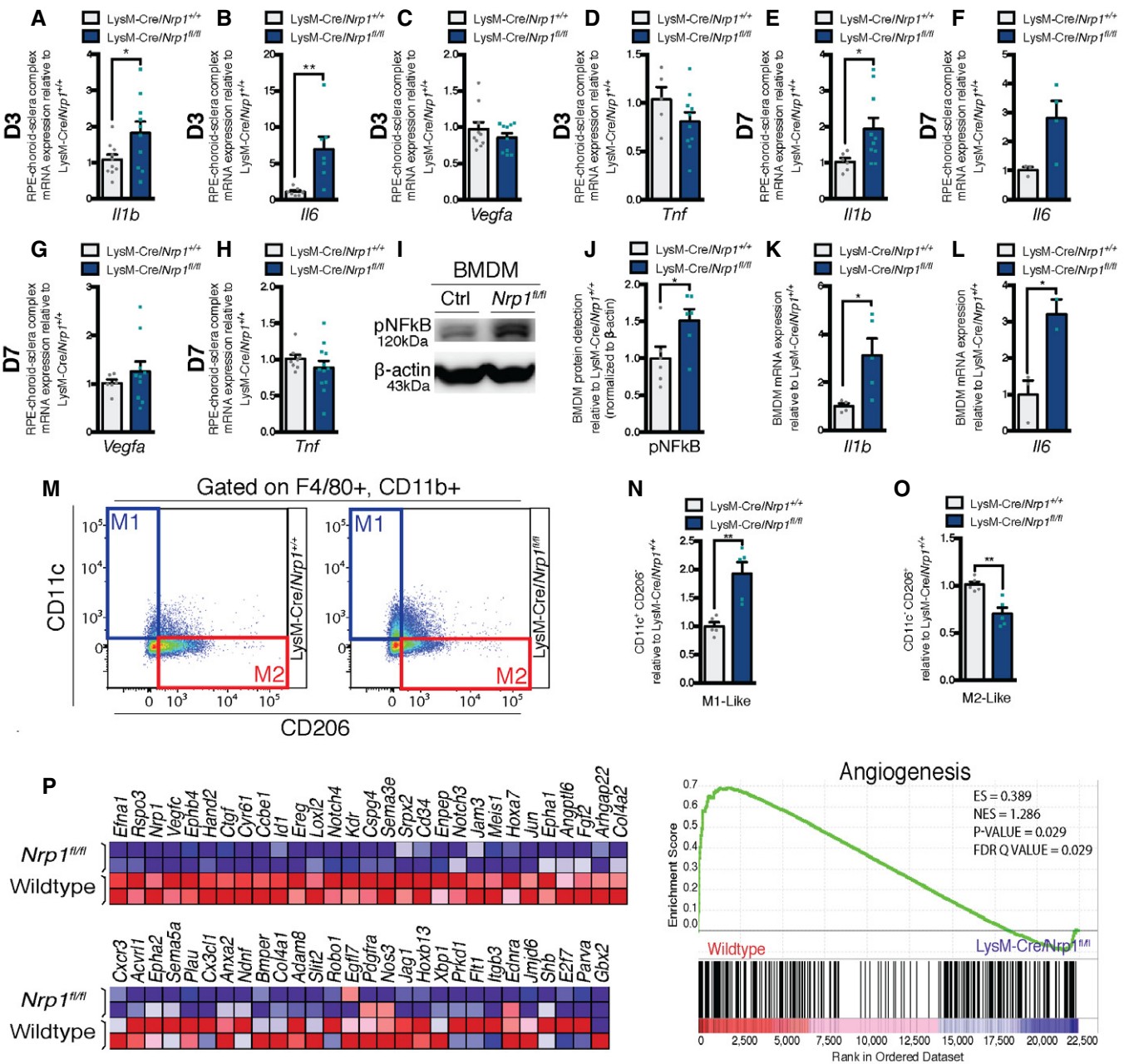

**Figure 3.**

similar for both ligands, SEMA3A showed a much slower dissociation to the immobilized trap compared with VEGF-A, suggesting that the SEMA3A-trap complex is very stable.

Intravitreal injections of NRP1-derived trap in C57BL/6J wild-type mice at D0 led to a robust ~47% decrease in Dextran-FITC-perfused neovessels compared with vehicle controls at day 14 post-laser burn (Fig 4E and F). The average size of IB4-labeled impact area was not significantly affected by the treatment (Fig 4E and G) and ratios of FITC-labeled neovessels to the size of IB4-labeled post-laser burn scarring were reduced by 41% after trap treatment (Fig 4E and H). Transcripts for *Vegfa* and *Tnf* did not vary following treatment with trap while *Il6* rose (Fig EV3A–C).

In order to determine whether the beneficial effects of NRP1-derived traps on CNV were mediated by influencing NRP1+ mononuclear phagocytes, we injected traps into the vitreous of LysM-Cre/*Nrp1*^fl/fl^ and LysM-Cre/*Nrp1*^+/+^ mice following laser burn. Similar to wild-type mice, trap treatment in LysM-Cre/*Nrp1*^+/+^ led to a significant decrease in FITC-perfused vessels and FITC/IB4-ratios (Fig 4I, J and L). Treatment of LysM-Cre/*Nrp1*^fl/fl^ mice reduced CNV to similar levels as seen in controls; however, the magnitude of effect was diminished (Fig 4I–L). NRP1 is expressed by several other cells in the retina and sclera-choroid-RPE complex such as endothelial cells and neurons. Intravitreal injection of the NRP1-derived traps will influence ligands that signal in all NRP1-expressing cells.

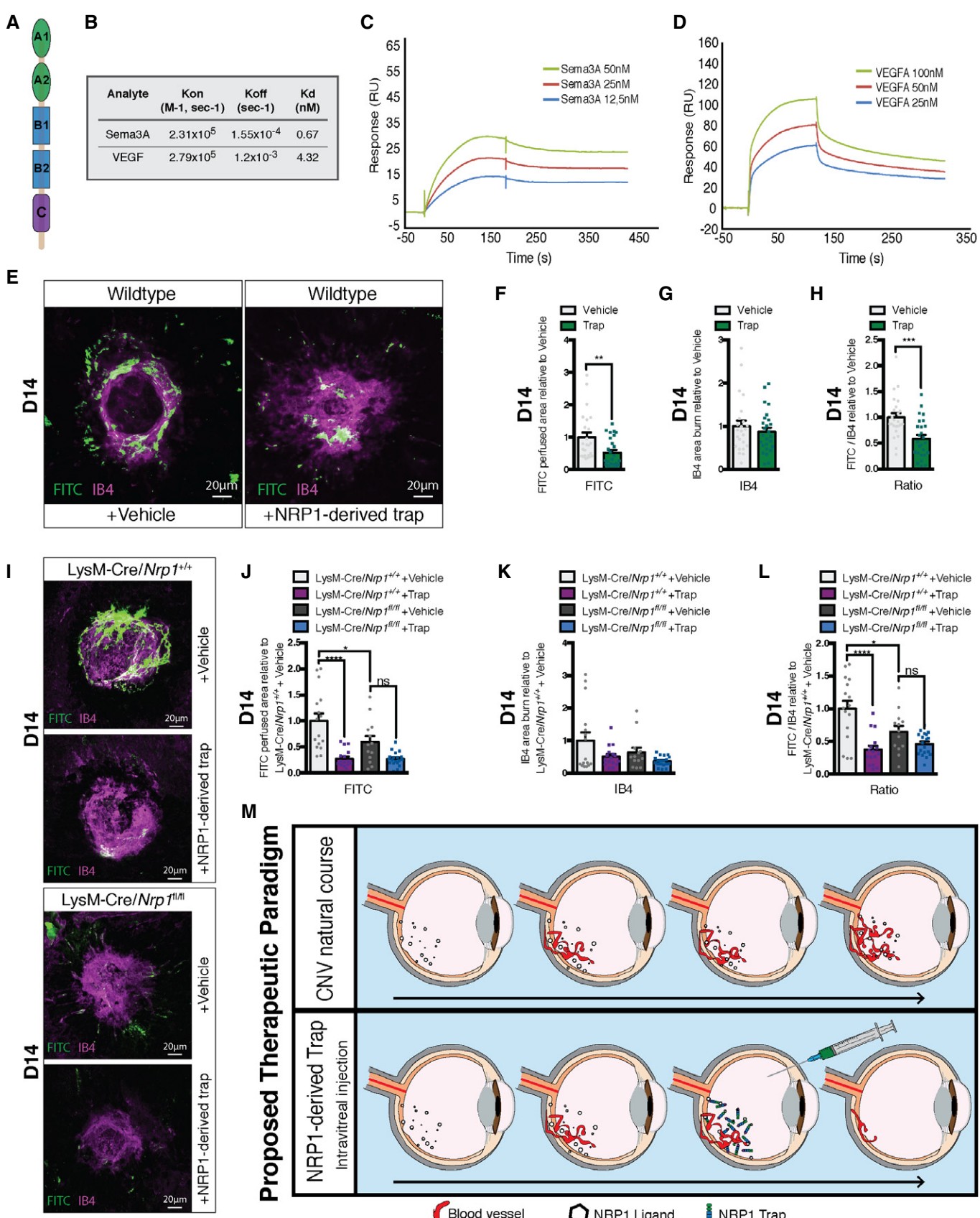

Figure 4.

**Figure 4. Therapeutic intravitreal administration of soluble NRP1 reduces CNV in mice.**

A   Schematic representation of a soluble receptor Neuropilin-1, consisting of five domains: two CUB motifs (A1, A2), two coagulation factor domains (B1, B2), and the MAM domain (C).
B   Rate constant and binding affinities of Sema3A and VEGF to immobilized trap obtained using a one-site Langmuir binding model.
C, D   Representative SPR sensorgrams for various concentrations of Sema3A (C) and VEGF (D) binding to immobilized Trap.
E   Compilation of representative compressed Z-stack confocal images of FITC–dextran-labeled CNV and isolectin B4 (IB4)-stained laser impact area from vehicle and NRP1-derived trap treated wild-type mice. Scale bar: 20 μm.
F–H   Quantification of area of FITC–dextran-labeled CNV (F), IB4-stained laser impact area (G) and the ratio of FITC/IB4 per laser burn (H) relative to vehicle at D14; $n = 24$ burns (vehicle), $n = 26$ burns (NRP1-derived trap).
I   Compilation of representative compressed Z-stack confocal images of FITC–dextran-labeled CNV and IB4-stained laser impact area from vehicle and NRP1-derived trap-treated LysM-Cre/*Nrp1*$^{+/+}$ and LysM-Cre/*Nrp1*$^{fl/fl}$ mice at D14. Scale bar: 20 μm.
J–L   Quantification of area of FITC–dextran-labeled CNV (J), isolectin B4 (IB4)-stained laser impact area (K), and the ratio of FITC/IB4 per laser burn (L) relative to LysM-Cre/*Nrp1*$^{+/+}$ + vehicle in vehicle and NRP1-derived trap-treated LysM-Cre/*Nrp1*$^{+/+}$ and LysM-Cre/*Nrp1*$^{fl/fl}$ mice at D14; $n = 16$ burns (LysM-Cre/*Nrp1*$^{+/+}$ + vehicle), $n = 16$ burns (LysM-Cre/*Nrp1*$^{+/+}$ + trap), $n = 13$ burns (LysM-Cre/*Nrp1*$^{fl/fl}$ + vehicle), $n = 19$ burns (LysM-Cre/*Nrp1*$^{fl/fl}$ + trap).
M   Proposed therapeutic paradigm. From a therapeutic perspective, intravitreal injection of NRP1-derived traps reduces pathological angiogenesis associated with CNV.

Data information: Comparisons between groups were analyzed using a Student's unpaired *t*-test; (E–G) or one-way ANOVA with Tukey's multiple comparisons test (I–K); *$P < 0.05$, **$P < 0.01$, ***$P < 0.001$, ****$P < 0.0001$; error bars represent mean ± SEM; exact $P$ values listed in Appendix Table S1.

The reduced therapeutic effect of the trap in LysM-Cre/*Nrp1*$^{fl/fl}$ mice suggests that a portion of the therapeutic effect of the NRP1 trap is mediated through myeloid cells. Together, these data suggest that sequestering NRP1 ligands is an effective strategy to reduce pathological subretinal neovascularization.

## Discussion

Here, we show that several ligands of NRP1 are induced in the vitreous of patients with active NV AMD and that myeloid-resident NRP1 contributes to pathological angiogenesis in later stages of CNV in mice. Moreover, we demonstrate that while mononuclear phagocyte-resident NRP1 is not essential for cellular recruitment to sites of CNV, it is critical for mitigating myeloid cell inflammation and skews myeloid cells toward a pro-angiogenic phenotype. The absence of NRP1 leads to enhanced production of pro-inflammatory factors as we have previously suggested for dendritic cells (Oussa *et al*, 2016) and adipose tissue macrophages (Wilson *et al*, 2018). These data add to the notion that less inflammatory and more M2-like mononuclear phagocytes are enriched with age, and exacerbate CNV in the laser-induced mouse model (Nakamura *et al*, 2015; Zandi *et al*, 2015).

Accumulation of mononuclear phagocytes in AMD secondary to the disruption of the physiologically immunosuppressive subretinal environment is central to the etiology of both atrophic and wet forms of disease (Sennlaub *et al*, 2013; Guillonneau *et al*, 2017; Rashid *et al*, 2019). Our data suggest that NRP1 keeps mononuclear phagocytes in a less inflammatory and more reparative state, toward the M2 portion of the spectrum. This is consistent with findings demonstrating that NRP1-deficient myeloid cells are more pro-inflammatory, classically activated cells in models of obesity (Wilson *et al*, 2018), tumor growth (Casazza *et al*, 2013; Roy *et al*, 2017; Miyauchi *et al*, 2018; Chen *et al*, 2019), and sepsis (Dai *et al*, 2017). With respect to being pro-angiogenic, NRP1-expressing mononuclear phagocytes have been described as dispensable for physiological angiogenesis in the retina and elsewhere (Fantin *et al*, 2013; Dejda *et al*, 2016), yet important for vessel growth during weight gain (Wilson *et al*, 2018). They have also been reported to normalize tumor blood vessels (Carrer *et al*, 2012) and promote pathological angiogenesis in the retina (Dejda *et al*, 2014) and tumors (Casazza *et al*, 2013).

Based on the above findings, we sought to determine the therapeutic potential of neutralizing NRP1 ligands. A single intravitreal injection of a recombinant NRP1-derived trap was effective at preventing CNV, highlighting the therapeutic potential of NRP1 for exudative AMD. In addition, others (Sodhi *et al*, 2019) and us (Cerani *et al*, 2013; Dejda *et al*, 2014) have demonstrated efficacy for recombinant NRP1 for retinal vasculopathies characterized by preretinal neovascularization or vasogenic edema. Potentially, non-responders to anti-VEGF therapy could benefit from this treatment paradigm, since a genetic variation of *Nrp1* is an indicator of reduced treatment response to anti-VEGF therapeutics like Ranibizumab in patients with NV AMD (Lores-Motta *et al*, 2016; Lores-Motta *et al*, 2018). NRP1-derived traps have a considerably lower affinity for VEGF-A compared with current anti-VEGF therapeutics such as Ranibizumab, Bevacizumab, and Aflibercept (García-Quintanilla *et al*, 2019) and hence may limit toxicity associated with sustained VEGF-A deprivation. Of note, the NRP1-derived trap was less effective in retinas of LysM-Cre/*Nrp1*$^{fl/fl}$ suggesting a mechanism of action in part redundant with deletion of NRP1 in myeloid cells.

In sum, while a role for endothelial-resident NRP1 has been demonstrated in choroidal and retinal neovascularization (Fernandez-Robredo *et al*, 2017), we provide evidence that NRP1-expressing immune cells contribute to CNV. Collectively, we provide rationale for therapeutic targeting of NRP1 ligands or NRP1-expressing myeloid cells for exudative AMD.

## Materials and Methods

### Vitrectomy

All patients were previously diagnosed with NV AMD and were followed and treated by a single vitreoretinal surgeon (F.A. Rezende). Control patients underwent surgical treatment for non-vascular pathology (epiretinal membrane or macular hole) by the same surgeon. Patients underwent surgery under local retro/peribulbar anesthesia. Three-port 25-gage transconjunctival pars plana vitrectomy was performed through 25-gage valved cannulas (Alcon). Under microscope visualization using a wide-angle viewing system (Resight, Zeiss), undiluted vitreous at the macular area was collected with a 25-gage vitrector. Vitreous samples were frozen

on dry ice immediately after biopsy and stored at −80°C. We obtained approval of human clinical protocols from the Hôpital Maisonneuve-Rosemont ethics committee (Ref. CER: 10059). Written informed consent was obtained from all subjects and the experiments conformed to the principles set out in the WMA Declaration of Helsinki and the Department of Health and Human Services Belmont Report.

## Quantification of ligands of NRP1 in human vitreous by ELISA

Samples were centrifuged at 15,000 *g* for 5 min at 4°C prior to analysis. NRP1 ligands were quantified in supernatants using ELISAs following manufacturer's instructions; VEGF-A (DVE00,. R&D Systems), SEMA3A (LS-F29822, LSBio), TGF-β (BMS-249-4, Thermo Fisher Scientific Inc.), PGF (EHPGF, Thermo Fisher Scientific Inc.), and PDGF-BB (BMS2071, Thermo Fisher Scientific Inc.)

## Animals

All studies were performed in accordance with the Association for Research in Visions and Ophthalmology (ARVO) Statement for the Use of Animals in Ophthalmic and Vision Research. All animal procedures were validated by the Animal Care committee of the University of Montreal and Hôpital Maisonneuve-Rosemont in agreement with the guidelines established by the Canadian Council on Animal Care.

C57BL/6J wild-type (WT), LysM-Cre (Lyz2tm1(cre)Ifo/J; no. 004781), and Neuropilin1-floxed (Nrp1tm2Ddg/J; no. 005247) mice were purchased from the Jackson Laboratory (Bar Harbor, ME, USA) and bred in house. We generated a line of myeloid-specific transgenic mice by breeding LysM-Cre mice (Cre-recombinase expressed in the myeloid lineage) with NRP1-floxed mice, resulting in a mouse with attenuated *Nrp1* in myeloid cells (LysM-Cre/*Nrp1*$^{fl/fl}$). Mice were raised under sterile barrier conditions and housed under a 12-h light cycle with water and food ad libitum. Only male mice were used in this study.

## *In vivo* imaging following laser-induced choroidal neovascularization (CNV)

*In vivo* imaging was performed using a scanning laser ophthalmoscope (Micron IV; Phoenix Laboratories, Pleasanton, CA, USA). Mice of 9–11 weeks of age were subjected to pupil dilation (Mydriacyl; Alcon, Mississauga, ON, Canada) and anesthetized with a mix of 10% ketamine and 4% xylazine (10 μl/g body weight). Fluorescein (Alcon, 1 unit/g body weight of a 5% fluorescein dilution in 0.9% sodium chloride) was injected subcutaneously and corneas were lubricated with Optixcare ophthalmic gel (Aventix Animal Health, Burlington, ON, Canada). After a fluorescein circulation of 5 min, retinas were imaged before and after inducing choroidal neovascularization with 4 distinct laser burns (50 μm, 300 mW, 0.05 s). Animals were followed-up 3, 7, and 14 days after laser burn.

## Surface plasmon resonance

Surface plasmon resonance analyses were performed using a Biacore T200 instrument (GE Healthcare). Purified recombinant trap was immobilized by standard amine-coupling chemistry on a Biacore CM4 carboxymethylated dextran sensor chip, which was pre-activated with 100 mM N-hydroxysuccinimide (NHS) and 100 mM of 3-(N,N-dimethylamino) propyl-N-ethylcarbodiimide (EDC). Surfaces were blocked by injecting 1 M ethanolamine. An immobilization abundance of 100–150 RU of Trap was reached. SEMA3A and VEGF-A were, respectively, injected over the sample and reference flow cells at increasing concentrations (12.5–100 nM) at a flow rate of 40 μl/min in PBS buffer supplemented with 0.025% (v/v) Tween-20. Binding sensorgrams were obtained by subtracting the reference flow cell. Response curves were analyzed using BIAevaluation software (GE Healthcare), and data from all concentrations were globally fit to a one-site Langmuir binding model.

## Real-time quantitative PCR analysis

Immediately after enucleation, eyes were dissected to isolate the sclera–choroid–RPE cell complex. BMDMs were washed 3× in PBS and collected in TRIzol. RNA was isolated using TRIzol and digested with DNase I to prevent amplification of genomic DNA contaminants. All-In-One RT MasterMix (ABM) was used for the reverse transcription, and BrightGreen qPCR MasterMix (ABM) to determine gene expression in an ABI Biosystems Real-Time PCR machine with β-actin (Actb) as a reference gene. We used the following primers: Mouse *Actb* = F: 5'-GAC GGC CAG GTC ATC ACT ATT G-3', R: 5'-CCA CAG GAT TCC ATA CCC AAG A-3'; Mouse *Vegfa* = F: 5'-GCC CTG AGT CAA GAG GAC AG-3', R: 5'-CTC CTA GGC CCC TCA GAA GT-3'; Mouse *Sema3a* = F: 5'-GGG ACT TCG CTA TCT TCA GAA-3', R: 5'-GGC GTG CTT TTA GGA ATG TTG-3'; Mouse *Tgfb1* = F: 5'-ACG CCT GAG TGG CTG TCT TTT GAC-3', R: 5'-GGG CTG ATC CCG TTG ATT TCC ACG-3'; Mouse *Pdgfb* = F: 5'-GAA GTT GGC ATT GGT GCG AT-3', R: 5'-TGG AGT CGA GTC GGA AAG CT-3'; Mouse *Nrp1* = F: 5'- ACC CAC ATT TCG ATT GGA G-3', R: 5'-TTC ATA GCG GAT GGA AAA CC-3'; Mouse *Il1b* = F: 5'-CTG GTA CAT CAG CAC CTC ACA-3', R: 5'-GAG CTC CTT AAC ATG CCC TG-3'; Mouse *Il6* = F: 5'-AGA CAA AGC CAG AGT CCT TCA GAG A-3', R: 5'-GCC ACT CCT TCT GTG ACT CCA GC-3'; Mouse *Tnf* = F: 5'-CCC TCA CAC TCA GAT CAT CTT CT-3', R: 5'-GCT ACG ACG TGG GCT ACA G-3'; Mouse *iNos* = F: 5'-CGG CAA ACA TGA CTT CAG GC-3', R: 5'-GCA CAT CAA AGC GGC CAT AG-3'; Mouse *Cd163* = F: 5'-ATG CTT CCA TCC AGT GCC TC-3', R: 5'-CAC AAA CCA AGA GTG CCG TG-3'; Mouse *Cd206* = F: 5'-GTT CAC CTG GAG TGA TGG TTC TC-3', R: 5'-AGG ACA TGC AGG GTC ACT T-3'; Mouse *Arg1* = F: 5'-CAF CAC TGA GGA AAG CTG GT-3', R: 5'-CAG ACC GTG GGT TCT TCA CA-3'; Mouse *Pgf* = F: 5'-CAG TTG CTT CTT ACA GGT CC-3', R: 5'-CAC CTC ATC AGG GTA TTC AT-3'.

## Laser-induced CNV

At the age of 6 weeks, mice were anesthetized by intraperitoneal injection with 10 μl/g body weight of a 10% ketamine and 4% xylazine solution. Using an argon laser, we ruptured their Bruch's membrane, as described previously (Lambert *et al*, 2013). Mice were sacrificed at 3, 7, 10, or 14 days after we induced 4 burns per eye for the immunohistochemistry analysis and 6 burns per eye for the RT-qPCR and FACS analyses.

## Immunohistochemistry

7 or 14 days after CNV induction, mice were sedated with isoflurane gas and cardiacally perfused with 0.5 ml of 15 mg/ml of fluorescein isothiocyanate (FITC)-dextran (average mol wt 2,000 kDa) and euthanized. Eyes were enucleated and fixed for 30 min in 4% PFA at room temperature, before dissection of the sclera-choroid-RPE cell complex. After a secondary fixation of 15 min in 4% PFA at room temperature, the choroids were stained with rhodamine-labeled Griffonia (bandeiraea) Simplicifolia Isolectin I (RL-1102-2, Vector Laboratories Inc.) (1:100), NRP1 (AF566, goat polyclonal; R&D Systems) (1:250), and IBA-1 (019-19741, rabbit polyclonal; Wako) (1:350) overnight. After 1-h incubation with secondary antibodies, the sclera-choroid-RPE cell complex was mounted onto a slide, and the burns and macrophages were captured in a Z-stack with an Olympus FV1000 microscope. The Z-stacks were compressed into one image and quantified in ImageJ.

## FACS on retina and sclera-choroid-RPE cell complexes

Retinas and sclera-choroid-RPE cell complexes of non-burned (D0) and burned mice at D3 were cut into small pieces and homogenized in a solution of 750 U/ml DNAse I (Sigma-Aldrich Corp.) and 0.5 mg/ml of collagenase (Roche) for 20 min at 37°C. Homogenates were filtered through a 70 μm cell strainer and washed in PBS. Viability of the cells was checked by Zombie Aqua (423101: BioLegend) staining for 15 min at room temperature. After incubation with LEAF-purified anti-mouse CD16/32 (101310; BioLegend) for 10 min at 4°C to block Fc receptors, cells were incubated for 25 min at 4°C with the following antibodies: BV711 anti-mouse/human CD11b (101242; BioLegend), PE anti-mouse F4/80 (123110; BioLegend), APC anti-mouse CD64 (139305; BioLegend), FITC anti-mouse CD38 (102705; BioLegend), APC/Cy7 anti-mouse Ly-6G (127624; BioLegend), BV785 anti-mouse CD11c (117335; BioLegend), and PE/Cy7 anti-mouse CD206 (141719; BioLegend). Antibody dilution was determined with titration by lot. Fluorescence-activated cell sorting (FACS) was performed on a BD LSRFortessaTM X-20 cell analyzer, and data were analyzed using FlowJo software (FlowJo version 10.2).

## Generation of BMDM

Bone marrow from both femurs and tibiae was harvested through bone flushing with PBS supplemented with 10% FBS. After red blood cell (RBC) lysis, cells were seeded in complete medium (Dulbecco's modified Eagle's medium (DMEM) plus 10% FBS and 1% streptomycin/penicillin) and stimulated with macrophage colony-stimulating factor (M-CSF) (Mouse M-CSF Recombinant Protein, eBioscience™; Invitrogen) 1:5,000. After 3 days of incubation at 37°C with 5% $CO_2$, fresh medium containing M-CSF was added. Cells were allowed to differentiate for a total of 6 days, before their medium was replaced by complete medium without M-CSF. LPS stimulated cells were stimulated for 24 h with 250 ng/ml LPS (*Escherichia coli* O55:B5 lipopolysaccharide; Sigma-Aldrich). The cells were not tested for mycoplasma contamination. As evaluated by flow cytometry, the purity was usually around 99%.

## Western blot analysis

For assessment of BMDM protein levels, we collected BMDM by scraping the cells in 1× RIPA on ice. Protein concentration was assessed by bicinchoninic acid (BCA) assay (Sigma-Aldrich), and 30 μg protein was analyzed for each condition by standard SDS–PAGE technique. Anti-NRP1 antibody (ab81321) (1:2,000) and Anti-NF-κB p65 (ab16502) (1:500) antibody were purchased from Abcam.

## FACS on BMDM for extracellular staining

BMDMs were collected in PBS through scraping. Viability of the cells was checked by Zombie Aqua (423101: BioLegend) staining for 15 min at room temperature. After incubation with LEAF-purified anti-mouse CD16/32 (101310; BioLegend) for 10 min at 4°C to block Fc receptors, cells were incubated for 25 min at 4°C with the following antibodies: BV711 anti-mouse/human CD11b (101242; BioLegend), PE anti-mouse F4/80 (123110; BioLegend), APC anti-mouse CD64 (139305; BioLegend), FITC anti-mouse CD38 (102705; BioLegend), APC/Cy7 anti-mouse Ly-6G (127624; BioLegend), BV785 anti-mouse CD11c (117335; BioLegend), and PE/Cy7 anti-mouse CD206 (141719; BioLegend). Antibody dilution was determined with titration by lot. Fluorescence-activated cell sorting (FACS) was performed on a BD LSRFortessaTM X-20 cell analyzer, and data were analyzed using FlowJo software (FlowJo version 10.2).

## RNA-seq sample preparation, sequencing, and analysis

RNA-seq data are available from a previous study (Wilson *et al*, 2018) in Gene Expression Omnibus (GEO) under the entry GSE110447. RNA-seq was performed as described previously (Wilson *et al*, 2018). Gene set enrichment analysis GSEA was conducted using GSEA v2.2.1 software provided by Broad Institute of Massachusetts Institute of Technology and Harvard University. We used the ANGIOGENESIS gene set contributed by the Gene Ontology Consortium from the Molecular Signature Database of the Broad Institute, Inc.

## Intravitreal injections

Wild-type, LysM-Cre/*Nrp1*^fl/fl^, and LysM-Cre/*Nrp1*^+/+^ mice subjected to laser burn were intravitreally injected with NRP1-derived trap or vehicle (Saline 0.9%) on the day of the burn and sacrificed at D14.

## Statistical analysis

Data are presented as mean ± SEM Student's *t*-test was used to compare two different groups or, when indicated, a one-way analysis of variance (ANOVA) and Dunnett's multiple comparisons test. A $P < 0.05$ was considered statistically different. *N* was indicated for each experiment.

# Data availability

This study includes no data deposited in external repositories.

**Expanded View** for this article is available online.

## The paper explained

### Problem

Age-related macular degeneration (AMD) is a slowly progressing condition of the aging eye and the leading cause of central vision loss in industrialized countries. Advanced AMD is often classified into "dry" atrophic AMD or "wet" neovascular (NV) AMD. Wet AMD typically occurs when neovascularization from the choroid (choroidal neovascularization; CNV), sprouts into the subretinal space and neuro-retina, hemorrhages, leaks and ultimately provokes photoreceptor death, fibrovascular scarring, and retinal detachment of the macular region. This can rapidly compromise the central visual field.

### Results

In the current study, we show that several ligands of Neuropilin1 (NRP1, a transmembrane receptor that binds several growth factors and guidance cues and potentiates their signaling) are induced in the vitreous of patients with active NV AMD. We observed that myeloid cells expressing NRP1 contribute to pathological angiogenesis in later stages of CNV in mice. Moreover, we demonstrate that while mononuclear phagocyte-resident NRP1 is not essential for recruitment of immune cells to sites of CNV, it is critical for mitigating myeloid cell inflammation and favors alternative activation. Using a NRP1-derived trap, we significantly reduced CNV.

### Impact

Our study ultimately shows that therapeutic targeting of NRP1 ligands or NRP1-expressing myeloid cells hinders CNV.

## Acknowledgments

We dedicate this study to the memory of our colleague and friend Normand Beaulieu who made invaluable contributions to the understanding of Neuropilin biology and the translational merit of targeting its ligands.
P.S. holds the Wolfe Professorship in Translational Research and a Canada Research Chair in Retinal Cell Biology. This work was supported by operating grants to P.S from The Foundation Fighting Blindness Canada and SemaThera Inc. Additional funding was provided by the Canadian Institutes of Health Research (Foundation grant #148460), the Diabetes Canada (DI-3-18-5444-PS), Heart and Stroke Foundation of Canada (G-16-00014658), and Natural Sciences and Engineering Research Council of Canada (418637), the Fonds de Recherche en Ophtalmologie de l'Université de Montréal (FROUM), and the Réseau en Recherche en Santé de la Vision. M.H. holds the Banting Fellowship from the CIHR. S.C-G. holds a Fonds de Recherche Santé du Québec (FRQS) scholarship.

## Author contributions

Research design and study: E.A., P.S.; Experimental work: E.A., F.B., F.F., F.P., M.H., A.D., G.M., A.W., N.B., K.B., S.C-G., S.B.; All retinal surgeries: F.A.R.; Data analysis: E.A., F.B., A.W., M.B.; Valuable conceptual insight on research design: JS.D., G.C., V.B.; Manuscript writing with valuable input from authors: E.A., A.W., P.S.

## Conflict of interest

P.S. is the founder of and a consultant for SemaThera Inc. G.C. and V.B. are consultants for SemaThera Inc. F.B., N.B., and K.B. are employees of SemaThera Inc. The rest of the authors declare that they have no conflict of interest.

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
