## [Review Process File · EMBO Molecular Medicine]

Myeloid-resident Neuropilin-1 promotes Choroidal Neovascularization while Mitigating Inflammation

Elisabeth Andriessen, Francois Binet, Frederik Fournier, Masayuki Hata, Agnieszka Dejda, Gaelle Mawambo, Frederique Pilon, Flavio Rezende, Karine Beauchemin, Sergio Crespo-Garcia, Garth Cumberlidge, Véronique Bougie, Ariel Wilson, Manuel Buscarlet, Steve Bourgault, Normand Beaulieu, Flavio Rezende, Jean-Sébastien Delisle, and Przemyslaw Sapieha

DOI: 10.15252/emmm.201911754

Corresponding author: Przemyslaw Sapieha (mike.sapieha@umontreal.ca)

Review Timeline:

Submission Date:	13th Nov 19
Editorial Decision:	6th Dec 19
Revision Received:	8th Oct 20
Editorial Decision:	30th Oct 20
Revision Received:	12th Jan 21
Editorial Decision:	18th Jan 21
Revision Received:	11th Feb 21
Accepted:	24th Feb 21

Editor: Lise Roth

Transaction Report:

6th Dec 2019

Dear Prof. Sapieha,

Thank you for the submission of your manuscript to EMBO Molecular Medicine. We have now received feedback from the three reviewers who agreed to evaluate your manuscript. As you will see from the reports below, the referees acknowledge the interest of the study and are overall supporting publication of your work pending appropriate revisions.

Addressing the reviewers' concerns in full will be necessary for further considering the manuscript in our journal, and acceptance of the manuscript will entail a second round of review. EMBO Molecular Medicine encourages a single round of revision only and therefore, acceptance or rejection of the manuscript will depend on the completeness of your responses included in the next, final version of the manuscript. For this reason, and to save you from any frustrations in the end, I would strongly advise against returning an incomplete revision.

When submitting your revised manuscript, please carefully review the instructions that follow below. Failure to include requested items will delay the evaluation of your revision:

- 1) A .docx formatted version of the manuscript text (including legends for main figures, EV figures and tables). Please make sure that the changes are highlighted to be clearly visible.
- 2) Individual production quality figure files as .eps, .tif, .jpg (one file per figure).
- 3) A .docx formatted letter INCLUDING the reviewers' reports and your detailed point-by-point responses to their comments. As part of the EMBO Press transparent editorial process, the point-by-point response is part of the Review Process File (RPF), which will be published alongside your paper.
- 4) A complete author checklist, which you can download from our author guidelines (<https://www.embopress.org/page/journal/17574684/authorguide#submissionofrevisions>). Please insert information in the checklist that is also reflected in the manuscript. The completed author checklist will also be part of the RPF.
- 5) Before submitting your revision, primary datasets produced in this study need to be deposited in an appropriate public database (see <https://www.embopress.org/page/journal/17574684/authorguide#dataavailability>). Please remember to provide a reviewer password if the datasets are not yet public. The accession numbers and database should be listed in a formal "Data Availability" section (placed after Materials & Method). Please note that the Data Availability Section is restricted to new primary data that are part of this study.

- 6) We would also encourage you to include the source data for figure panels that show essential

data. Numerical data should be provided as individual .xls or .csv files (including a tab describing the data). For blots or microscopy, uncropped images should be submitted (using a zip archive if multiple images need to be supplied for one panel). Additional information on source data and instruction on how to label the files are available at .

7) Our journal encourages inclusion of *data citations in the reference list* to directly cite datasets that were re-used and obtained from public databases. Data citations in the article text are distinct from normal bibliographical citations and should directly link to the database records from which the data can be accessed. In the main text, data citations are formatted as follows: "Data ref: Smith et al, 2001" or "Data ref: NCBI Sequence Read Archive PRJNA342805, 2017". In the Reference list, data citations must be labeled with "[DATASET]". A data reference must provide the database name, accession number/identifiers and a resolvable link to the landing page from which the data can be accessed at the end of the reference. Further instructions are available at .

8) We replaced Supplementary Information with Expanded View (EV) Figures and Tables that are collapsible/expandable online. A maximum of 5 EV Figures can be typeset. EV Figures should be cited as 'Figure EV1, Figure EV2' etc... in the text and their respective legends should be included in the main text after the legends of regular figures.

- Additional Tables/Datasets should be labeled and referred to as Table EV1, Dataset EV1, etc. Legends have to be provided in a separate tab in case of .xls files. Alternatively, the legend can be supplied as a separate text file (README) and zipped together with the Table/Dataset file. See detailed instructions here:

9) The paper explained: EMBO Molecular Medicine articles are accompanied by a summary of the articles to emphasize the major findings in the paper and their medical implications for the non-specialist reader. Please provide a draft summary of your article highlighting

10) For more information: There is space at the end of each article to list relevant web links for further consultation by our readers. Could you identify some relevant ones and provide such information as well? Some examples are patient associations, relevant databases, OMIM/proteins/genes links, author's websites, etc...

11) Every published paper now includes a 'Synopsis' to further enhance discoverability. Synopses are displayed on the journal webpage and are freely accessible to all readers. They include a short stand first (maximum of 300 characters, including space) as well as 2-5 one-sentences bullet points that summarizes the paper. Please write the bullet points to summarize the key NEW findings.

They should be designed to be complementary to the abstract - i.e. not repeat the same text. We encourage inclusion of key acronyms and quantitative information (maximum of 30 words / bullet point). Please use the passive voice. Please attach these in a separate file or send them by email, we will incorporate them accordingly.

Thank you for providing a synopsis image. Please make sure the text is readable when the image is resized to a jpeg file 550 px-wide x 400-px high.

12) A Conflict of Interest statement should be provided in the main text

13) As part of the EMBO Publications transparent editorial process initiative (see our Editorial at <http://embomolmed.embopress.org/content/2/9/329>), EMBO Molecular Medicine will publish online a Review Process File (RPF) to accompany accepted manuscripts.

In the event of acceptance, this file will be published in conjunction with your paper and will include the anonymous referee reports, your point-by-point response and all pertinent correspondence relating to the manuscript. Let us know whether you agree with the publication of the RPF and as here, if you want to remove or not any figures from it prior to publication.

EMBO Molecular Medicine has a "scooping protection" policy, whereby similar findings that are published by others during review or revision are not a criterion for rejection. Should you decide to submit a revised version, I do ask that you get in touch after six months if you have not completed it, to update us on the status.

I look forward to receiving your revised manuscript.

Sincerely,

Lise Roth

Lise Roth, PhD
Editor
EMBO Molecular Medicine

To submit your manuscript, please follow this link:

Link Not Available

***** Reviewer's comments *****

Referee #1 (Comments on Novelty/Model System for Author):

State of the art experimental models to study retinal immunity and choroidal neovascularization. The depth of data analysis from RNA Seq analyses could be improved, see comments.

Referee #1 (Remarks for Author):

Andriessen et al present a very interesting short report on the role of neuropilin-1 in retinal mononuclear phagocytes and the effects in the laser-induced model of choroidal neovascularization. As key findings, the group showed that cell-specific knockout in phagocytes or injection of NRP1 trap molecules significantly reduced late stage phagocyte recruitment and CNV formation. These data are especially interesting as further angiogenesis-related molecules in addition to VEGF could be targets to treat neovascular retinal diseases such as age-related macular degeneration of diabetic macular edema.

The paper is well written and many conclusions are supported by the data. The findings fit very well in the scope of EMM and should not only be important for vision research but also for angiogenesis researcher in general.

I have a few minor comments that could improve the quality of the paper further:

1. Introduction and following text. I understand that the total word count for reports is limited but the abbreviation MP's for mononuclear phagocytes is not very elegant. I would recommend to use the terms "mononuclear phagocytes", "phagocytes", "macrophages" etc. where appropriate at the relevant text passages.
2. Fig. 1 shows ELISA data for VEGF, SEMA3A, TGF β and PDGF β in ocular fluids from patients/controls and mRNA data from the laser CNV model. It could be interesting to include placental growth factor (PGF) in the mRNA analyses as recent data in the same mouse model showed a prominent PGF expression in phagocytes and a differential regulation by anti-VEGF injections. It would be nice to see whether NRP1 has an influence on PGF levels.
3. Figure 2 H etc. I understand that the absolute numbers and Iba1+ areas are not significantly different in NRP1^{-/-} vs NRP1^{+/+} laser spots at day 7. I suggest that the authors could reanalyze their flat mount image data using grid cross analyses or other tools to determine the ramification changes as potential early events before cell numbers may change at later time points.
4. Figure 3 A-H denotes choroidal mRNA expression. I wonder whether the RPE/choroid complex (it is hard to dissect RPE from choroidal tissue) or indeed the choroid alone was analyzed here. If the first assumption is the case, the Y-axis labels should be changed to RPE/choroid.
5. The group presents preliminary bioinformatic analyses from NRP1 deficient and control peritoneal macrophages. I would suggest a) to deposit the raw data in a repository such as GEO and b) to perform a more detailed analysis of the transcriptomic changes to better define the proposed phagocyte polarization. I noticed that the authors were very carefully handling the M1/M2 macrophage paradigm, which is indeed a problematic topic when analyzing tissues and not cells, but the more sophisticated analysis of RNAseq may provide further insights into the cellular events in the absence or presence of NRP1.

Referee #2 (Remarks for Author):

The present study from Przemyslaw Sapieha's lab investigated the function of Nrp1+ macrophages in the progression of the human AMD and mouse CNV. The experiments using LysM-Cre mediated Nrp1 deletion in mice revealed that Nrp1 expressed on macrophages promotes perfusion of pathologically expanding neovessels. Interestingly the Nrp1-Trap injection showed therapeutic effects on such perfusion. Overall, the data promote our understanding of retinal neovascular diseases and may lead to new therapeutic options. However, this reviewer find several points needed to be addressed as listed below:

1. (Fig. 1A) In this reviewer's opinion, this scheme is unnecessary as we find everywhere.
2. (Fig. 2G) In the panels of LysM-Cre/Nrp1+/+, Nrp1 does not merge with Iba1 in some places (upper right and upper left area). As such immunoreactivity diminished in CKO mice, it may not be non-specific. Could authors comment on this unexplained staining?
3. (Fig. 2O, Fig 4B) What is the FITC+IB4- substances? Does not IB4 stain all blood vessels in this model? Supple Fig. 1D looks differently. Also, the FITC positive cells in Fig. 4B do not look like ECs. In any case, the section immunostaining or orthogonal view might be helpful to demonstrate FITC is indeed located in the endothelial lumen.
4. The section title "NRP1-expressing MP's display a pro-angiogenic alternatively activated phenotype" is not correct. Perhaps it should be "NRP1-deficient MP...".
5. The molecular mechanism how loss of Nrp1 leads to M1 polarization is totally unclear. In this reviewer's opinion, authors should address by additional experiments investigating the intra-cellular signaling, or discuss much referring to the relevant literature.
6. The data do not support "Graphical Summary", in particular "proliferation" and "fibrosis" have not been substantially analyzed. The scheme should reflect the actual data, but not speculation from the expression profiles.

Referee #3 (Remarks for Author):

NRP1 binds several pro-angiogenic ligands in association with various co-receptors. In this manuscript, Andriessen et al. describe a pro-angiogenic role for NRP1+ retinal myeloid cells in a model of wet age-related macular degeneration (AMD). The authors show that NRP1-expressing phagocytes, which express a pro-angiogenic/alternatively-activated phenotype, accumulate in the retina after laser-induced damage in the Brunch membrane and therein promote choroidal neovascularization (CNV), which ultimately results in AMD. In LysM.Cre/Nrp1^{fl/fl} mice, myeloid cell-specific inactivation of Nrp1 skews the polarization of retinal myeloid cells toward a more pronounced pro-inflammatory phenotype and reduces CNV. In order to determine the therapeutic potential of blocking the interaction between NRP1 and its ligands (including VEGFA), the authors used an NRP1-derived extracellular trap. When injected in the vitreus, the NRP1-derived trap reduced the laser-induced CNV area. Because the NRP1 trap can potentially block a number of pro-angiogenic ligands, the authors conclude that this approach may represent an effective strategy to

reduce pathological CNV also in wet AMD patients who fail on VEGFA blockade.

The study is overall well performed. However, I have some concerns with conceptual flow, translational advance, as well as mechanistic insight.

Mechanistic insight:

1. A premise of the study is that NRP1 binds several pro-angiogenic factors; so, interfering with NRP1 may potentially block multiple pro-angiogenic pathways in wet AMD. The authors should investigate whether the NRP1-derived extracellular trap indeed sequesters VEGFA along with other pro-angiogenic factors. Is SEMA3A neutralized?
2. More extensive characterization of the retinas of mice that received the NRP1 trap should be provided. Indeed, the only readout used here is CNV. What about IL1B and IL6 levels? Also, it is unclear whether NRP1-expressing myeloid cells were affected.
3. How does Nrp1 deletion in myeloid cells enhance mRNA expression of Il1b and Il6 in the injured tissue? Is this increase cell-autonomous in the Nrp1-deficient myeloid cells or non-cell-autonomous in other cells of the tissue? Do mRNA levels correlate with protein levels?
4. Related to point 3 above: Do increased IL1B and IL6 have a role in limiting CNV? Blocking experiments were not performed.
5. More generally, there is little compelling evidence for NRP1 to control the M1-M2 myeloid cell phenotype in the retina. Most of the studies were conducted on marrow-derived macrophages.

Translational advance:

6. Related to point 1 above, the trap should be compared with a pure VEGFA inhibitor.
7. Related to point 1 above: Should the trap be found to mainly sequester VEGFA, what would the advantage of using it as apposed to standard-of-care VEGFA/VEGFR2 blockade?
8. The novelty of the study is partly limited by previous work. For example, the presence of VEGFA in the vitreous of AMD patients (e.g. Hsu et al, Scientific reports, 2016 PMID:27725716) and the presence of SEMA3 in the context of neovascular eye diseases (e.g. Bai et al, Molecular Vision, 2014 PMID:25352735) have already been reported previously. Also, there is abundant literature on NRP1-expressing myeloid cells and their pro-angiogenic functions.

Conceptual flow:

9. It is unclear (at least to me) what the NRP1 trap experiment was performed. Indeed, the first part of the study focused on the role of NRP1 in myeloid cell-mediated CNV, whereas the NRP1 trap experiment addresses the potential role of NRP1 ligands in CNV. Because NRP1 is expressed by vascular endothelial cells and, potentially, other cell types, the NRP1 trap experiment does not in any way address the contribution of NRP1-expressing myeloid cells to CNV. The NRP1 trap should also be tested in mice with Nrp1-deficient myeloid cells in order to formulate a relevant mechanistic

hypothesis.

Other points:

10: The proficiency of Nrp1 deletion should also be tested in microglial cells and not only in BMDMs.

11. The authors refer to Table 1, which was not present in this version of the manuscript.

12. The gating strategy in Supplemental Figure 1A lacks of crucial controls. As the main message of this paper is based on the presence of NRP1+ cells, the authors should provide control flow cytometry dot plots (FMOs in particular). The NRP1+ population presented in this figure appears questionable and not convincing based on the data shown.

13: Figures 3P and 3Q are not described in the text.

Detailed response to reviewers

Reviewer #1:

We thank the reviewer for their thoughtful comments and positive assessment of our study and thank them for acknowledging that this is a 'very interesting short report that uses state of the art experimental models'. We would also like to thank them for suggesting that "the findings should not only be important for vision research but also for angiogenesis researcher in general." Based on the recommendations and queries of the reviewer, we have performed a series of new experiments that have permitted us to provide additional data.

Query 1: Introduction and following text. I understand that the total word count for reports is limited but the abbreviation MP's for mononuclear phagocytes is not very elegant. I would recommend to use the terms "mononuclear phagocytes", "phagocytes", "macrophages" etc. where appropriate at the relevant text passages.

Answer: We thank the reviewer for this suggestion and we agree with them. We have made all the requested changes.

Query 2: Fig. 1 shows ELISA data for VEGF, SEMA3A, TGFb and PDGFb in ocular fluids from patients/controls and mRNA data from the laser CNV model. It could be interesting to include placental growth factor (PGF) in the mRNA analyses as recent data in the same mouse model showed a prominent PGF expression in phagocytes and a differential regulation by anti-VEGF injections. It would be nice to see whether NRP1 has an influence on PGF levels.

Answer: This is indeed an interesting addition to figure 1, since the b1 and b2 domains of NRP1 can bind PGF. We have therefore conducted an ELISA for PGF on the human vitreous samples from patients with NV-AMD and qPCRs with a pgf primer on the RPE-choro-sclera complexes of mice at different timepoints. The ELISA showed no difference in expression of PGF between patients with NV-AMD and control group. The mouse choroids showed a dip in pgf expression 3 days after laser burn. The pgf timecourse follows roughly the same path as Crespo et al in "Inhibition of Placenta Growth Factor Reduces Subretinal Mononuclear

Phagocyte Accumulation in Choroidal Neovascularization” published in IOVS in 2017, with the important difference that we do not have the timepoint D1 and therefore did not detect the increase in expression seen at D1.

Query 3: Figure 2 H etc. I understand that the absolute numbers and Iba1+ areas are not significantly different in NRP1^{-/-} vs NRP1^{+/+} laser spots at day 7. I suggest that the authors could reanalyze their flat mount image data using grid cross analyses or other tools to determine the ramification changes as potential early events before cell numbers may change at later time points.

Answer: We thank the reviewer for this suggestion and we have now performed grid-cross analysis and findings are provided to the reviewer. We found morphological data consistent with mononuclear phagocytes in LysM-Cre/Nrp1^{fl/fl} mice initially more ramified and then slightly less suggesting a state resembling M2 and then M1. While interesting, we believe that we would need to further characterize the state of these cells on a molecular levels to draw definitive conclusions. This is unfortunately not possible in the current frame of the study.

Figure for reviewers removed.

Query 4: Figure 3 A-H denotes choroidal mRNA expression. I wonder whether the RPE/choroid complex (it is hard to dissect RPE from choroidal tissue) or indeed the choroid alone was analyzed here. If the first assumption is the case, the Y-axis labels should be changed to RPE/choroid.

Answer: The reviewer is correct. We have made changes to the labeling of the graphs.

Query 5: The group presents preliminary bioinformatic analyses from NRP1 deficient and control peritoneal macrophages. I would suggest a) to deposit the raw data in a repository such as GEO and b) to perform a more detailed analysis.

Answer: We agree with the reviewer and we have deposited the transcriptomic files into GEO Database as GSE110447. Furthermore, in figure 3Z, we now provide an individual gene expression data as a heat map.

Reviewer #2

We are grateful to the reviewer for their insightful evaluation, elegant suggestions and positive appraisal of our study. We thank them for acknowledging that the “data promote our understanding of retinal neovascular diseases and may lead to new therapeutic options”. The suggestions of the reviewer have led to several additional experiments and we believe we have addressed their concerns.

Query 1: 1. (Fig. 1A) In this reviewer's opinion, this scheme is unnecessary as we find everywhere.

Answer: Thank you for this comment. We have now removed the scheme of NRP1 from figure 1A.

Query 2: (Fig. 2G) In the panels of LysM-Cre/Nrp1+/+, Nrp1 does not merge with Iba1 in some places (upper right and upper left area). As such immunoreactivity diminished in CKO mice, it may not be non-specific. Could authors comment on this unexplained staining?

Answer: The reviewer is right and we have changed the images. We must add that NRP1 is only knocked out on the Lys expressing cells and as a consequence mice still express NRP1 on other cells of non myeloid origin. The light staining of NRP1 in LysM-Cre/Nrp1fl/fl is a consequence of the presence of other NRP1 expressing cells, likely endothelial and neuronal.

Query 3: (Fig. 2O, Fig 4B) What is the FITC+IB4- substances? Does not IB4 stain all blood vessels in this model? Supple Fig. 1D looks differently. Also, the FITC positive cells in Fig. 4B do not look like ECs. In any case, the section immunostaining or orthogonal view might be helpful to demonstrate FITC is indeed located in the endothelial lumen.

Answer: We thank the reviewer for this comment. We have used the FITC perfusion model

where we inject a dextran-FITC solution into the left arterial chamber before we sacrifice the mouse. This allows for dextran-FITC to perfuse the neovessels. Isolectin IB₄ stains not only endothelial cells but also certain immune cells, and some neurons. The underlying choroid is often visible after staining with IB₄ because the laser impact has damaged all RPE cells in the center of the burn, exposing IB₄ positive choroid and scar tissue. In our experience IB₄-staining gives a gross overestimation of the neovascularization. We used FITC as an additional technique to measure CNV. Since only vessels with a lumen will be perfused with FITC the IB₄ staining is also necessary. We use the ratio of IB₄/FITC to give an idea of both the laser impact (IB₄ positive) and the formation of CNV (dextran-FITC perfused).

Query 4: The section title "NRP1-expressing MP's display a pro-angiogenic alternatively activated phenotype" is not correct. Perhaps it should be "NRP1-deficient MP...?"

Answer: We believe the data in the section suggests that NRP1 on mononuclear phagocytes renders them pro-angiogenic and in line with an alternatively activated phenotype.

Query 5: The molecular mechanism how loss of Nrp1 leads to M1 polarization is totally unclear. In this reviewer's opinion, authors should address by additional experiments investigating the intra-cellular signaling, or discuss much referring to the relevant literature.

Answer: We agree. We have in the current study limited this section to demonstrating that NRP1 deficiency leads to Nf-kB p65 activation as well as production of cytokines typically associated with M1 such as IL-1b and Tnf-a

Query 6: The data do not support "Graphical Summary", in particular "proliferation" and "fibrosis" have not been substantially analyzed. The scheme should reflect the actual data, but not speculation from the expression profiles.

Answer: We agree with the reviewer and have modified the diagram .

Reviewer #3

We thank the reviewer for their perceptive, very helpful and pertinent comments. We also thank them for acknowledging that "The study is overall well performed ". We have conducted several new experiments in line with the reviewer's comments and believe they have overall strengthened the manuscript.

Query 1: A premise of the study is that NRP1 binds several pro-angiogenic factors; so, interfering with NRP1 may potentially block multiple pro-angiogenic pathways in wet AMD. The authors should investigate whether the NRP1-derived extracellular trap indeed sequesters VEGFA along with other pro-angiogenic factors. Is SEMA3A neutralized?

Answer: We thank the reviewer for this comment. We have now performed Surface Plasmon Resonance (SPR) assessment of the NRP1-derived traps to VEGF and SEMA3A. These data are now shown in Figure 4 B-D.

Query 2: More extensive characterization of the retinas of mice that received the NRP1 trap should be provided. Indeed, the only readout used here is CNV. Also, it is unclear whether NRP1-expressing myeloid cells were affected.

Answer: We have now quantified VEGFA, IL6 and TNFa levels at D3 after NRP1-derived trap injection. These data are presented in Supplemental Figure 3. Essentially, we only noticed IL6 to rise which may be part of a remodeling mechanism.

Query 3: How does Nrp1 deletion in myeloid cells enhance mRNA expression of Il1b and Il6 in the injured tissue? Is this increase cell-autonomous in the Nrp1-deficient myeloid cells or non-cell-autonomous in other cells of the tissue? Do mRNA levels correlate with protein levels?

Answer: We provide evidence in the revised version of Figure 3 that NRP1-deficient MP's remain "locked" in an M1-like phase and show an increased inflammatory response. In BMDMs we now show that lack of NRP1 activates Nf-Kb p65 and that they have significantly elevated baseline production of IL1 and IL6. Since the culture of BMDM was >99% pure and there were no other cell types present, we conclude that it's a cell autonomous process in the NRP1-deficient myeloid cells.

Query 4: More generally, there is little compelling evidence for NRP1 to control the M1-M2 myeloid cell phenotype in the retina. Most of the studies were conducted on marrow-derived macrophages.

Answer: We agree with the reviewer have removed these claims and significantly de-emphasized the polarization aspect of the study.

Query 5: Related to point 1 above: Should the trap be found to mainly sequester VEGFA, what would the advantage of using it as apposed to standard-of-care VEGFA/VEGFR2 blockade?

Answer: We find the traps do sequester VEGFA (Figure 4B-D) however with a higher Kd than current anti-VEGF. Given that the traps are administered at 1µg/ul and there is a significant stoichiometric excess and hence we would not expect an additive effect. Due to lockdown measures, we were unable to prioritize the experiment.

Query 6: The novelty of the study is partly limited by previous work. For example, the presence of VEGFA in the vitreous of AMD patients (e.g. Hsu et al, Scientific reports, 2016 PMID:27725716) and the presence of SEMA3 in the context of neovascular eye diseases (e.g. Bai et al, Molecular Vision, 2014 PMID:25352735) have already been reported previously. Also, there is abundant literaturature on NRP1-expressing myeloid cells and their pro-angiogenic functions.

Answer: We thank the reviewer for this comment. The novelty of our study lies in the fact that we demonstrate that NRP1 on myeloid cells prevents an the cells from triggering release of proinflammatory cytokines. Furthermore, we link heightened inflammation due to absence of NRP1 on myeloid cells to lower CNV. Lastly we provide a therapeutic entity (NRP1-derived trap) for CNV. Hence we believe all 3 points provide novelty.

Query 7: It is unclear (at least to me) what the NRP1 trap experiment was performed. Indeed, the first part of the study focused on the role of NRP1 in myeloid cell-mediated CNV, whereas the NRP1 trap experiment addresses the potential role of NRP1 ligands in CNV. Because NRP1 is expressed by vascular endothelial cells and, potentially, other cell types, the NRP1 trap experiment does not in any way address the contribution of NRP1-expressing myeloid cells to CNV. The NRP1 trap should also be tested in mice with Nrp1-deficient myeloid cells in order to formulate a relevant mechanistic hypothesis.

Answer: We have now tested the NRP-1-derived trap in LysM-Cre/Nrp1fl/fl mice. The trap suppressed CNV to levels observed in C57BL6 mice. However, the LysM-Cre/Nrp1fl/fl already show significantly reduced levels of CNV. Hence the magnitude of reduction of CNV by NRP1-derived traps in LysM-Cre/Nrp1fl/fl was not statistically significant when compared to C57BL6 mice suggesting convergent mechanisms of action.

Query 8: The proficiency of Nrp1 deletion should also be tested in microglial cells and not only in BMDMs.

Answer: We have previously published this data in PMID: 27035626 and found that roughly 30% of retinal microglia are depleted for NRP1 in this model.

Query 9: The authors refer to Table 1, which was not present in this version of the manuscript.

Answer: We thank the reviewer for this comment. We had omitted to submit the table. It is now included in the submission.

Query 10: The gating strategy in Supplemental Figure 1A lacks of crucial controls. As the main message of this paper is based on the presence of NRP1+ cells, the authors should provide control flow cytometry dot plots (FMOs in particular). The NRP1+ population presented in this figure appears questionable and not convincing based on the data shown.

Answer: Thank you for this comment. We conducted these experiments and published the NRP1 signal and isotype control in supplemental S4 panel B of Wilson et al. (PMID: 29549139).

Query 11: Figures 3P and 3Q are not described in the text.

Answer: We thank the reviewer for bringing this to our attention. We have added the description for this section.

30th Oct 2020

Dear Prof. Sapieha,

Thank you for the submission of your revised manuscript to EMBO Molecular Medicine. We have now received feedback from the two reviewers who were asked to re-evaluate your manuscript. As you will see from the reports below, both referees acknowledge your efforts to address their initial concerns, and recognize that the manuscript has significantly improved. However, referee #2 also mentions issues that remain unanswered and finds additional experiments and further discussion necessary to support the claims.

Therefore, we would like you to address this referee's concerns in a revised version of your manuscript. Please be aware that this will be the last chance for you to address these points.

When submitting your revised manuscript, please carefully review the instructions that follow below. Failure to include requested items will delay the evaluation of your revision:

- 1) A .docx formatted version of the manuscript text (including legends for main figures, EV figures and tables). Please make sure that the changes are highlighted to be clearly visible.
- 2) Individual production quality figure files as .eps, .tif, .jpg (one file per figure).
- 3) A .docx formatted letter INCLUDING the reviewers' reports and your detailed point-by-point responses to their comments. As part of the EMBO Press transparent editorial process, the point-by-point response is part of the Review Process File (RPF), which will be published alongside your paper.
- 4) We can accommodate a maximum of 5 keywords, please adjust accordingly.
- 5) Please make sure that all co-authors are entered in the submission system (Sergio Crespo-Garcia, Veronique Bougie, Steve Bourgault, Flavio A Rezende currently missing).
- 6) Author contributions: Frederique Pilon, Manuel Buscarlet, Steve Bourgault are not mentioned, please complete.
- 7) Please complete the list of funders in the submission system.
- 8) Please reformat the references so as to have 10 authors listed before et al.
- 9) Please make sure that all figures and figures panels are referenced in the main text.
- 10) Before submitting your revision, primary datasets produced in this study need to be deposited in an appropriate public database. The accession numbers and database should be listed in a formal "Data Availability" section (placed after Materials & Method). Please note that the Data Availability Section is restricted to new primary data that are part of this study. Alternatively, please indicate: "This study includes no data deposited in external repositories".
- 11) We would also encourage you to include the source data for figure panels that show essential

data. Numerical data should be provided as individual .xls or .csv files (including a tab describing the data). For blots or microscopy, uncropped images should be submitted (using a zip archive if multiple images need to be supplied for one panel). Additional information on source data and instruction on how to label the files are available at

12) Please indicate in the figure legends if Fig 2N was reused in Fig 4H.

13) Appendix: please combine the three appendix figures as one PDF file including the legends (the appendix figure legends should be removed from the main manuscript and added to the appendix file, together with a table of content). The nomenclature needs to be corrected in the figure legends.

14) The paper explained: EMBO Molecular Medicine articles are accompanied by a summary of the articles to emphasize the major findings in the paper and their medical implications for the non-specialist reader. Please provide a draft summary of your article highlighting

15) For more information: There is space at the end of each article to list relevant web links for further consultation by our readers. Could you identify some relevant ones and provide such information as well? Some examples are patient associations, relevant databases, OMIM/proteins/genes links, author's websites, etc...

16) Every published paper now includes a 'Synopsis' to further enhance discoverability. Synopses are displayed on the journal webpage and are freely accessible to all readers. They include a short stand first (maximum of 300 characters, including space) as well as 2-5 one-sentences bullet points that summarizes the paper. Please write the bullet points to summarize the key NEW findings. They should be designed to be complementary to the abstract - i.e. not repeat the same text. We encourage inclusion of key acronyms and quantitative information (maximum of 30 words / bullet point). Please use the passive voice. Please attach these in a separate file or send them by email, we will incorporate them accordingly.

We note that you provided 3 images as synopsis. Are some of them suggestions for cover?

17) As part of the EMBO Publications transparent editorial process initiative (see our Editorial at <http://embomolmed.embopress.org/content/2/9/329>), EMBO Molecular Medicine will publish online a Review Process File (RPF) to accompany accepted manuscripts.

In the event of acceptance, this file will be published in conjunction with your paper and will include the anonymous referee reports, your point-by-point response and all pertinent correspondence relating to the manuscript. Let us know whether you agree with the publication of the RPF and as here, if you want to remove or not any figures from it prior to publication.

EMBO Molecular Medicine has a "scooping protection" policy, whereby similar findings that are published by others during review or revision are not a criterion for rejection. Should you decide to

submit a revised version, I do ask that you get in touch after three months if you have not completed it, to update us on the status.

I look forward to receiving your revised manuscript.

Yours sincerely,

Lise Roth

Lise Roth, PhD
Editor
EMBO Molecular Medicine

***** Reviewer's comments *****

Referee #2 (Remarks for Author):

In this revised paper, authors have adequately addressed my previous concerns and strengthened the data. Now the paper is acceptable.

Referee #3 (Remarks for Author):

The authors have addressed some of the initial concerns of this reviewer. However, my general concerns with conceptual flow, translational advance, as well as mechanistic insight, unfortunately remain. The authors have provided very brief responses to the reviewer's comments (generally a couple of sentences), often without describing and explaining new data and their significance.

In general, I remain confused about the conclusions that can be drawn from the NRP1 deletion studies and the NRP1 trap studies, as the two approaches target different mechanisms. The author's response to point 7 is not clear and the conclusion that there are "convergent mechanisms of action" is not compelling. Rather, the data in Fig 4I seem to support the notion that different

mechanisms (and targets) are involved. In fact, the trap seems to decrease CNV in the NRP1-deficient setting, although the data are not statistically different. (lack of statistical significance can be due to high variation and low numerosity, not necessarily to the lack of biological effects.)

Points 8 and 10 should be addressed with pertinent new data (key controls), because results may differ from those obtained in previous studies. Also, it is concerning that NRP1 was deleted in only about 30% of the retinal microglia (response to point 8). This point should be discussed.

The corresponding author is listed as affiliated to (employed by?) SemaThera, but in the COI section they are listed as consultant. The COI of the corresponding author should be better defined.

Even with the key limitations mentioned above, the ms could be published after the remaining points have been addressed and discussed in the paper.

Itemized list of corrections**Queries from October 30th****Referee #2:**

Query 1: In this revised paper, authors have adequately addressed my previous concerns and strengthened the data. Now the paper is acceptable.

Answer: We thank the reviewer for their time and for providing helpful comments that have altogether improved the quality of the study.

Referee #3:

Query 1: In general, I remain confused about the conclusions that can be drawn from the NRP1 deletion studies and the NRP1 trap studies, as the two approaches target different mechanisms. Conclusion that there are "convergent mechanisms of action" is not compelling. Rather, the data in Fig 4I seem to support the notion that different mechanisms (and targets) are involved.

Answer: We agree with the reviewer. We have modified the text to include the notion that only part of the effects of the trap are mediated through myeloid cells. Moreover, we explain that NRP1 is expressed by several other cells in the retina and sclera-choroid-RPE complex. This suggests that a portion of the therapeutic effect of the NRP1 trap is mediated through myeloid cells. We have modified and clarified this in the revised text to include these changes.

Query 2: NRP1 was deleted in only about 30% of the retinal microglia. This point should be discussed.

Answer: We agree with the reviewer and have added data to the supplemental section of the text. Concerning expression levels of NRP1 in myeloid cells and microglia, we had obtained data during this study but not yet added in this manuscript, since we performed these experiments for the purpose of assuring ourselves that our results are in line with those obtained in previous studies from our lab (PMID: 27035626 and PMID: 29549139). They are now included in the supplementals. We have not performed the experiments with publication of the results in mind and the number of animals we sacrificed for this experiment were low yet the number of cells we obtained in our FACS samples were sufficient enough to confirm

the same values we previously obtained in our publications. Both in naïve choro-RPE-retina complexes and 3 days after burn we find an ~27% reduction in NRP1 levels in microglia (see below). ISO and FMO control graphs that were used in the analysis have also been included.

Query 3: The corresponding author is listed as affiliated to (employed by?) SemaThera, but in the COI section they are listed as consultant. The COI of the corresponding author should be better defined.

Answer: The corresponding author is the founder of and a consultant for SemaThera Inc. This is now clearly stated in the manuscript.

Editorial Queries:

The authors performed the requested editorial changes.

Queries from December 6th

Referee #3:

Query 1: The data should be included as a supplemental figure, as it is essential to show the extent of gene deletion for those crucial studies; the data should also be discussed. Gene deletion was apparently quite low, as shown by the minimal decrease of NRP1+ microglia. The reader should be aware of this limitation of the study and of its implications for the interpretation of the results.

Answer: As suggested, we have now added the requested data showing the extent of gene deletion to revised manuscript. We also now discuss these data in the manuscript.

Editorial Queries:

Query 2: Please check that the funding section in the submission system matches the funders listed in the manuscript. There is at least one that is not in the system.

Answer: We have now included all funding sources in the submission system.

18th Jan 2021

Dear Prof. Sapieha,

Thank you for the submission of your revised manuscript to EMBO Molecular Medicine, and please accept my apologies for the delay in getting back to you, which is due to the limited staff and increased submitted manuscripts during the holiday season. We have now received the enclosed reports from referee #3 who re-reviewed your manuscript. As you will see, this referee is supportive of publication, and we will be able to accept your manuscript pending the following final minor amendments:

1) Main manuscript text:

- Please answer/correct the changes suggested by our data editors in the main manuscript file (in track changes mode). This file will be sent to you in the next couple of days. Please use this file for any further modification.
- Please remove the colored text, and only keep in track changes any new modification.
- Material and Methods:
 - o Patients samples: Please identify the committee(s) approving the study protocol. Please include the full statement that informed consent was obtained from all subjects and the experiments conformed to the principles set out in the WMA Declaration of Helsinki and the Department of Health and Human Services Belmont Report.
 - o Cells: please indicate whether the cells were tested for mycoplasma contamination.
 - o Mice: Please provide the housing and husbandry conditions. Please indicate the gender of the mice used in your experiments.
 - o Antibodies: please provide antibody dilutions for all experiments.
 - o RNA sequencing: it is unclear whether you performed RNAseq in this study or analysed data from previous study (Wilson et al, 2018). Please clarify.
- Data Availability Section: Please note that the Data Availability Section is restricted to new primary data that are part of this study. The accession number you provided links to datasets generated in a previous study (Wilson et al, 2018). Please clarify. If new experiments were performed, please provide a link to the public repository where data have been deposited. If no new primary dataset was produced, please indicate "This study includes no data deposited in external repositories"
- Please indicate in the figures or in the legends the exact n= and exact p= values, not a range, along with the statistical test used. Some people found that to keep the figures clear, providing a supplemental table with all exact p-values was preferable. You are welcome to do this if you want to.
- Please remove the Graphical Summary
- Please replace "Competing interests" by "Conflict of interest"
- Thank you for updating the reference format, however, we note that some references are still incorrect. Please update all references so as to have 10 authors listed before et al.

2) Checklist:

Section C/7: Please indicate whether the cells were tested for mycoplasma contamination.

Section D/8 and 9: please provide the requested information (including housing and husbandry conditions).

Section E: Please fill in section 11 and 12.

Section F/18: this section should list the accession codes for data generated in this study. The accession number provided refers to RNAseq data generated in a previous study (Wilson et al, 2018). Please clarify.

3) Thank you for providing a synopsis. I slightly edited it to match our style and format. Please let me know if you agree with the following:

A population of innate immune myeloid cells expressing the NRP1 receptor invades the retina and drives pathological neovascularization during age-related macular degeneration (AMD). A recombinant NRP1-derived trap prevents pathological angiogenesis associated with choroidal neovascularization.

- NRP1 ligands were elevated in patients with neovascular AMD and in a mouse model of choroidal neovascularization (CNV).
- NRP1-expressing mononuclear phagocytes rose in the retina upon injury and promoted CNV.
- CNV was reduced in mice by therapeutic intravitreal administration of soluble NRP1.

Thank you for providing a synopsis image. Would you agree with using only the upper part (Graphical Summary)? When combined with "Proposed Therapeutic Paradigm" and resized so as to have a width of 550px, the text becomes too small and the image a bit packed.

4) You indicated that you agree with the publication of the Review Process File (RPF) with your manuscript. Please let us know whether you want to remove or not any figures from it prior to publication.

I look forward to receiving your revised manuscript.

Yours sincerely,

Lise Roth

Lise Roth, PhD

Editor

EMBO Molecular Medicine

***** Reviewer's comments *****

Referee #3:

I checked the final minor revisions and I am satisfied.

The authors performed the requested editorial changes.

24th Feb 2021

Dear Prof. Sapieha,

Thank you for sending the revised files. I have looked at everything, and all is fine. I am therefore very pleased to accept your manuscript for publication in EMBO Molecular Medicine!

Please note that I changed the section F/18 of the checklist, as this concerns only newly generated dataset, and I therefore indicated: "This study includes no data deposited in external repositories." Please contact us immediately if you do not agree.

Your manuscript will be sent to our publisher to be included in the next available issue of EMBO Molecular Medicine.

Please read below for additional important information regarding your article, its publication and the production process.

Congratulations on a nice study!

Yours sincerely,

Lise Roth

Lise Roth, Ph.D
Editor
EMBO Molecular Medicine

Corresponding Author Name: Sapięha Przemysław

Manuscript Number: EMM-2019-11754